# A long-term dataset of simulated epilimnion and hypolimnion temperatures in 401 French lakes (1959-2020)

Najwa Sharaf[1,2], Jordi Prats[3], Nathalie Reynaud[1,2], Thierry Tormos[1,4], Rosalie Bruel[1,4], Tiphaine Peroux[1,2], Pierre-Alain Danis[1,4]

[1]Pôle R&D Ecosystèmes Lacustres (ECLA), OFB-INRAE-USMB, Aix-en-Provence, France
[2]INRAE, Aix Marseille Univ, RECOVER, Team FRESHCO, 3275 Route Cézanne, 13182 Aix-en-Provence, France
[3]SEGULA Technologies, C. Calàbria 169, 08015 Barcelona, Spain
[4]OFB, DRAS, Service ECOAQUA, 3275 Route Cézanne, 13100 Aix-en-Provence, France

*Correspondence to:* Sharaf Najwa (najwa.sharaf@inrae.fr), Bruel Rosalie (rosalie.bruel@ofb.gouv.fr), Tormos Thierry (thierry.tormos@ofb.gouv.fr), Reynaud Nathalie (nathalie.reynaud@inrae.fr)

## 1. Abstract

Understanding the thermal behavior of lakes is crucial for water quality management. Under climate change, lakes are warming and undergoing alterations in their thermal structure, including surface and deep-water temperatures. These changes require continuous monitoring due to the possible major ecological implications on water quality and lake processes. We combined numerical modelling and satellite thermal data to create a regional dataset (LakeTSim: Lake Temperature Simulations) of long-term water temperatures for 401 French lakes in order to tackle the scarcity of in situ water temperature. The dataset consists of daily epilimnion and hypolimnion water temperatures for the period 1959-2020 simulated with the semi-empirical OKPLM (Ottosson-Kettle-Prats Lake Model) and the associated uncertainties. Here, we describe the model and its performance. Additionally, we present the uncertainty analysis of simulations with default parameter values (parametrized as a function of lake characteristics) and calibrated parameter values, along with the analysis of the sensitivity of the model to parameter values and biases in the input data. Overall, the 90% confidence uncertainty range is largest for hypolimnion temperature simulations with a median of 8.5 °C and 2.32 °C respectively with default and calibrated parameter values. There is less uncertainty associated with epilimnion temperature simulations with a median of 5.42 °C and 1.85 °C, respectively before and after parameter calibration. This dataset provides over six decades of epilimnion and hypolimnion temperature data, crucial for climate change studies at a regional scale. It will help provide insight into the thermal functioning of French lakes and can be used to help decision-making and stakeholders.

## 2. Introduction

Lakes, both natural and artificial (i.e., reservoirs and gravel pits) are sentinels of environmental change and provide important services such as access to drinking water, hydropower production, recreation and fisheries (Adrian et al., 2009). Under climate change and anthropogenic pressures, many lakes are warming and consequently experiencing changes to their biophysicochemical structure and function that are leading to services being compromised (Janssen et al., 2021).

In lakes, water temperature is an essential parameter regulating processes such as the functioning of trophic webs, oxygen conditions, the physical structure of the water column as well as the biogeochemistry (Yang et al., 2018). Under warming, historical records and future projections demonstrate that for lakes, alterations in the thermodynamic functioning including warmer temperatures and shifts in mixing regimes already took place and are expected to persist in the future (Shatwell et al., 2019; Woolway and Merchant, 2019). In this context, they are undergoing shorter periods of ice cover and longer, more stable periods of thermal stratification (Woolway et al., 2022). These alterations could have considerable ecological implications for the biological communities (Lind et al., 2022; Havens and Jeppesen, 2018). For instance, worldwide studies have shown that the expansion of toxic cyanobacterial blooms is linked to warming (Griffith and Gobler, 2020). Other responses include species reduced body size (Daufresne et al., 2009), changes in thermal habitat and shifts in species seasonality (Kharouba et al., 2018).

It is thus crucial to closely evaluate water temperature trajectories over the entire water column in space and time when assessing the impact of climate change on lake ecosystems. However, the lack of data coverage, both spatially and temporally, makes it difficult to accurately characterize lakes thermal response to climate change and to identify warming trends (Gray et al., 2018). Indeed, long-term datasets of in situ temperatures are usually scarce and mostly limited to large lakes (Layden et al., 2015). Moreover, sampling frequency and temporal coverage of

in situ water temperature varies greatly from one lake to the next, from a few years (Sharma et al., 2015) up to
decades (Piccolroaz et al., 2020; Rimet et al., 2020).
Due to the difficulties in setting up conventional (i.e., in situ) monitoring programs tied to e.g., costs, governance
and intercalibration, the coupling of modelling and satellite remote sensing data has become fundamental in the
field of limnology (Nouchi et al., 2019). Modelling provides means to interpolate both temporal and spatial gaps.
It thereby allows us to acquire information about surface water temperatures, which are globally the focus of lake
climate change studies, and deep-water temperatures which are as critical though often disregarded in this context
(however see Pilla et al., 2020). Several numerical models that vary in complexity exist for conducting water
temperature simulations, the most accurate being deterministic or process-based models. Nevertheless, regional or
global deterministic modelling efforts over long periods are usually hindered by the lack of sufficiently detailed
input data (e.g., meteorological and field data) to run the models (Kim et al., 2021). For practical and operational
purposes, simpler models (semi-empirical, statistical or hybrid physical-statistical based models) with less
requirements for forcing data, have been mostly applied to assess the impact of climate change on lake ecosystems
and study them (Piccolroaz et al., 2020; Toffolon et al., 2014; Sharma et al., 2008). Long-term simulations across
a considerable number of lakes are made possible with this type of models, enabling the detection of trends in time
series data that are not achievable with shorter datasets (Gray et al., 2018).
The performance of numerical models depends highly on the calibration of their parameters as well as on the
quality of the input data. Satellite remote sensing is an effective way to monitor surface water temperature on a
synoptic scale (Schaeffer et al., 2018; Sharaf et al., 2019) and provide a complementary source of data to in situ
measurements for model calibration or validation purposes (Allan et al., 2016; Babbar-Sebens et al., 2013). In
particular, thermal infrared sensors onboard the Landsat satellites are very adequate for retrospective analysis of
surface water temperature with a spatial resolution adapted for small to medium size lakes and reservoirs at a
bimonthly acquisition frequency. Landsat 4 and 5 TM (Thematic Mapper), 7 ETM+ (Enhanced Thematic Mapper)
and 8 TIRS (Thermal InfraRed Sensor) provide surface temperature data at spatial resolutions of 120, 60 and 100
m respectively. Landsat series records of surface water temperature can be used to validate 3D hydrodynamic
models when in situ measurements are scarce (Sharaf et al., 2021) and to spatially assess the quality and suitability
of aquatic habitat for biological communities (Halverson et al., 2022). Although, satellite thermal data is limited
to the surface, its integration  into model calibration could improve the accuracy of simulations over the surface
layer and the water column (Javaheri et al., 2016).
Here we present on a regional scale, a long-term dataset, LakeTSim (Lake Temperature Simulations), of daily
epilimnion and hypolimnion temperature simulations, as well as uncertainties, for the period 1959-2020 over 401
French lakes monitored under the Water Framework Directive (WFD) including natural and artificial lakes,
reservoirs and gravel pits. We present the OKPLM (Ottosson-Kettle-Prats Lake Model) used to produce water
temperature simulations and its performance. Further, we provide the uncertainty analysis of simulations with
default (parametrized with in situ and satellite thermal data over an entire set of lakes) and calibrated (with in situ
temperature measurements for each lake) model parameter values as well as the sensitivity analysis for the latter.
The goal of publishing this dataset is to provide new insight about epi- and hypolimnion temperatures of lakes in
France especially for those that are not monitored regularly through conventional methods. This long-term dataset
is valuable for developing temperature indicators for identifying warming trends, extreme events and possible
changes in the mixing regime among others. These indicators will contribute to assess the impact of climate change
on lakes thermal functioning and its influence on the biological community structure and trophic webs.
**3. Data and methodology**
**3.1. The software suite ALAMODE**
The simulations, sensitivity and uncertainty analysis presented in this paper were made using the software suite
ALAMODE (A LAke MODElling project). ALAMODE (Danis, 2020) is a software suite developed in python 3
by the Pôle R&D Ecosystèmes Lacustres (ECLA) and SEGULA Technologies to facilitate the realization of
simulations of lakes and the management of related information. It comprises multiple modules and packages
designed for lake and tributary modelling, as well as for processing the data necessary to operate these models.
These packages include OKPLM (Ottosson-Kettle-Prats Lake Model), CUSPY (Calibration, Uncertainty analysis
and Sensitivity analysis in PYthon), TMOD (Temperature MODelling), GLMtools (General Lake Model tools),
"tributary", TINDIC (Temperature INDICators) and ALAPROD (ALAMODE-Production). OKPLM (Prats-
Rodríguez and Danis, 2023b) is used to simulate epilimnion and hypolimnion water temperatures in lakes while
CUSPY (Prats-Rodríguez and Danis, 2023a) is used for model parameters estimations and conducting uncertainty
and sensitivity analyses. TMOD is used for managing the T-MOD database designated to facilitate the realization
and consultation of simulations. GLMtools is used to conduct lake hydrodynamic simulations using the one-
dimensional hydrodynamic General Lake Model (Hipsey et al., 2019) while "tributary" is used for the estimation
of flow and temperature of lake tributaries. The package TINDIC is used for calculating temperature indicators
from model simulations. Finally, ALAPROD integrates all the functionalities to produce simulations into a single
package: simulation of stream water temperature, of lake hydrodynamics and temperature, and of stream flow rate.
It also includes sensitivity and uncertainty analysis features. The functionalities of these packages can be accessed
either by using each package separately or by utilizing the ALAPROD package, which depends on the TMOD
database and requires access to it.
At present, only the ALAMODE packages related to the main functionalities used in this work are publicly
available (see Code availability section): the simulation of lake temperatures using the Ottosson-Kettle-Prats Lake
Model (Prats & Danis, 2019), implemented in the package OKPLM, and the sensitivity and uncertainty analysis
tools in the CUSPY package. We used ALAPROD to access the functionalities of both packages.
**3.2. The OKP Lake Model description**
The OKPLM (Ottosson-Kettle-Prats Lake Model) (Prats & Danis, 2019) is a two-layer semi-empirical data model
adapted from Kettle et al (2004) for the epilimnion module and Ottosson & Abrahamsson (1998) for the
hypolimnion module. The modifications proposed in Prats & Danis (2019) consisted mainly of simplifying the
mixing algorithm used in Ottosson & Abrahamsson (1998) using a basic stability condition, whereas for the
epilimnion module a sinusoidal fit to average daily solar radiation was used instead of the theoretical clear-sky
radiation. The OKPLM also runs on weekly and monthly frequencies. The regionalization of the parameters of the
model mainly depends on the geographical and morphological properties of the lake (maximal depth, volume,
surface area, latitude and altitude). The model requires few meteorological forcing data: solar radiation and air
temperature.
The model calculates water temperature as follows:
$\quad T_{e,i} = A + Bf(T^*_{a,i}) + CS_i$ (1)
where $T_e$ is the epilimnion temperature (°C), $i$ is the day number, $A, B$ and $C$ are calibration parameters, $S$ is the
solar radiation (W m⁻²) and $f(*)$ is an exponential smoothing function with $T^*_{a,i}$ defined as:
$\quad T^*_{a,i} = T_{a,i} - MAAT$ (2)
Where $T_{a,i}$ is air temperature (°C) and $MAAT$ is the annual mean air temperature (°C). The smoothing function
$f(*)$ is such that it gives greater weight to the nearest observations and the weights decrease exponentially. It is
defined as:
$\quad f(T^*_{a,1}) = T^*_{a,1}$ (3)
$\quad f(T^*_{a,i}) = \alpha T^*_{a,i} + (1 - \alpha)f(T^*_{a,i-1})$ (4)
where $\alpha$ is the smoothing factor. When $\alpha = 1$ there is no smoothing, while the smoothing increases with the
decrease in the value of $\alpha$.
$\quad T_{h,i} = A \cdot D + E \cdot g(T_{e,i})$ (5)
where $T_h$ is the hypolimnion temperature (°C), $D$ and $E$ are calibration parameters and $g(T_{e,i})$ is an exponential
smoothing as follows:
$\quad g(T_{e,1}) = T_{e,1}$ (6)
$\quad g(T_{e,i}) = \beta T_{e,i} + (1 - \beta)g(T_{e,i-1})$ (7)
where $\beta$ is the exponential smoothing factor. As for $\alpha$, there is no smoothing for $\beta = 1$ and the smoothing increases
as $\beta$ approaches zero.
In ALAPROD, OKPLM can be run in two modes: the "default" mode where model parameter values for each
lake are estimated using the parameterization presented in Prats & Danis (2019), and the "calibrated" mode where
model parameters are calibrated individually for each lake by using in situ temperature measurements. The default
parameterization was obtained by using the individually calibrated parameter values to fit appropriate expressions
as a function of the characteristics of lakes. In the epilimnion module model parameter values $A$, $B$, $C$, and $\alpha$ are
estimated based on lake characteristics (i.e., latitude, altitude, surface area, volume, and depth). These equations
were determined using robust regressions and Landsat infrared data (median skin temperatures) from 1999 to 2016
of French lakes to estimate mean surface temperatures (Prats et al., 2018). In contrast, for the hypolimnion module,
parameter values $E$ and $\beta$ were derived as a function of lake depth and lake type using temperature profile data
from 357 lakes; $\beta$ can have a value of 1 ($E > 0.95$) or 0.13 ($E \leq 0.95$). The parameter $D$ was assigned a constant
value of 0.51.
The parametrization of the OKPLM parameters as presented in Prats & Danis (2019) is as follows:
$\quad A = 39.9 - 0.484L_{Lat} - 4.52 \times 10^{-3}L_{Alt} - 0.167lnL_A$ (8)
where $L_{Lat}$ is lake latitude (°N), $L_{Alt}$ is lake altitude (m) and $L_A$ is lake surface area (m²).
$$B = 1.058 - 0.0010L_{Dmax} \qquad (9)$$
where $L_{Dmax}$ is lake maximal depth (m).
$$C = 1.12 \times 10^{-3} - 3.62 \times 10^{-6}L_{Alt} \qquad (10)$$
$$E = e_1 + \frac{1-e_1}{1+\exp[e_3(e_2 - lnL_D)]} \qquad (11)$$
where $e_1$, $e_2$ and $e_3$ are coefficients with respective values of 0.10, 2.0, -1.8 for natural lakes and 0.49, 1.7, -2.0
for artificial lakes (reservoirs, gravel pits, ponds and quarry lakes) and $L_D$ is lake mean depth (m).
$$\alpha = \exp(0.52 - 3.0 \times 10^{-4}L_{Alt} + 0.25lnL_A - 0.36lnL_V) \qquad (12)$$
where $L_V$ is lake volume ($m^3$).

### 3.3.    Input data

The OKPLM was forced with two sources of meteorological data extracted from the SAFRAN (Système d'Analyse
Fournissant des Renseignements Adaptés à la Nivologie) analysis system (Durand et al., 1993) and the S2M
(SAFRAN–SURFEX/ISBA–Crocus–MEPRA) meteorological reanalysis (Vernay et al., 2015, 2022).
The SAFRAN system provides meteorological variables at an hourly time step estimated through interpolation
and assimilation processes with an 8 km square grid. Average daily data from the nearest grid cell was selected
for each study site. The difference in altitude between the study site and the grid cell was accounted for by applying
an adiabatic elevation correction on air temperature.
The S2M model chain combines the SAFRAN meteorological analysis and the SURFEX/ISBA–Crocus snow
cover model including MEPRA (Modèle Expert d'Aide à la Prévision du Risque d'Avalanche). It is more adapted
to mountainous regions as it has a spatial definition where spatial heterogeneity is taken into consideration. The
S2M reanalysis uses a vertical resolution of 300 m and is the result of simulations performed over mountainous
zones referred to as "massifs" and covering the French Alps, Pyrenees and Corsica mountainous regions. In order
to use S2M meteorological data over each lake an extraction of certain topographic classes is necessary. These
include elevation, aspect and slope, which represent the spatial variability over "massifs". On average, a massif
corresponds to a mountainous region of about 1000 $km^2$ over which meteorological conditions are considered
homogeneous at a given elevation range. Two types of S2M reanalysis simulations exist for each elevation range,
one at flat terrain and the other with 8 aspects at 2 different slope angles. For this study, this information (elevation,
slope, aspect) was extracted from a Digital Elevation Model (BD Alti, IGN) for each lake over its drainage basin,
combined into zones corresponding to S2M topographic classes.  We considered a zero slope and average daily
data for each study site.
In situ temperature profiles, geographical and morphological data of the study sites were initially extracted from
the PLAN_DEAU database. The extracted data was then incorporated into the T-MOD database, with the aim of
simplifying the process of simulations and accessing information about the characteristics of the simulated lakes.
Both databases are managed by INRAE (l'Institut National de Recherche pour l'Agriculture, l'Alimentation et
l'Environnement) and Pôle R&D ECLA ("ECosystèmes Lacustres") in Aix-en-Provence, France.   The
geomorphological data consisting of maximal depth, volume, surface area, latitude and altitude were extracted for
401 lakes. In situ temperature profiles were extracted for 170 lakes over different depths. Depending on each lake,
the number of years with samples could vary from 1 to 12 with a number of samples ranging between 1 and 10 per
year.
### 3.4.    Lake simulations
For this study, we considered 401 lakes (Figure 1) located in Metropolitan France monitored according to the
Water Framework Directive (WFD). Here we refer to lakes as natural lakes, reservoirs, gravel pits and other
artificial lakes (e.g., ponds and quarry lakes). The present lake dataset includes epi- and hypolimnion temperature
simulations for 54 natural lakes, 302 reservoirs, 7 gravel pits and 38 other artificial lakes (Figure 2). The lakes
characteristics range between 0 and 2279.7 m for altitude, 0.8 and 309.7 m for maximal depth, 0.08 and 577.12 km$^2$
for surface area and $5 \times 10^4$ and $8.9 \times 10^{10}$ m$^3$ for volume.
The OKPLM was run for each lake using either "default" or "calibrated" parameters, and either SAFRAN or S2M
meteorological data. Specifically, "calibrated" model parameters were adopted when in situ temperature profiles
along the water columns were available from the RCS/RCO (Réseau de Contrôle de Surveillance/Réseau de
Contrôle Opérationel, French networks for WFD) monitoring; these temperature profiles were then transformed
and used as epilimnion and hypolimnion temperatures. For those lakes, calibration parameters ($A$, $B$, $C$, $D$, $E$, $\alpha$
and $\beta$) are lake-specific and determined using the lake-specific temperature profiles. Conversely, "default"
parameters were used for the rest of the lakes as well as when bathymetry data necessary for the transformation of
temperature profiles into epilimnion and hypolimnion temperatures were not available. In this case, the values of
the parameters were estimated according to equations (8) to (12).
SAFRAN data were used for most of the lakes except for a few lakes at higher altitudes. Indeed, S2M data are
more representative of mountainous meteorological conditions than SAFRAN data and were thus used, when
possible, for simulating the water temperature in lakes situated at altitudes higher than 900 m. For some lakes, it
was not possible to use S2M data, either because their drainage basins are not entirely part of a massif ($n = 1$), or
because they are located in massifs that are not covered by the S2M reanalysis dataset ($n = 18$). Among the total
number of study sites ($n = 401$), the model was forced using SAFRAN and S2M meteorological data respectively
for 210 and 21 lakes with "default" model parameters, and for 164 and 6 lakes with "calibrated" model parameters.
The geomorphological characteristics of the simulated lakes with each of the abovementioned configurations are
shown in Table 1.

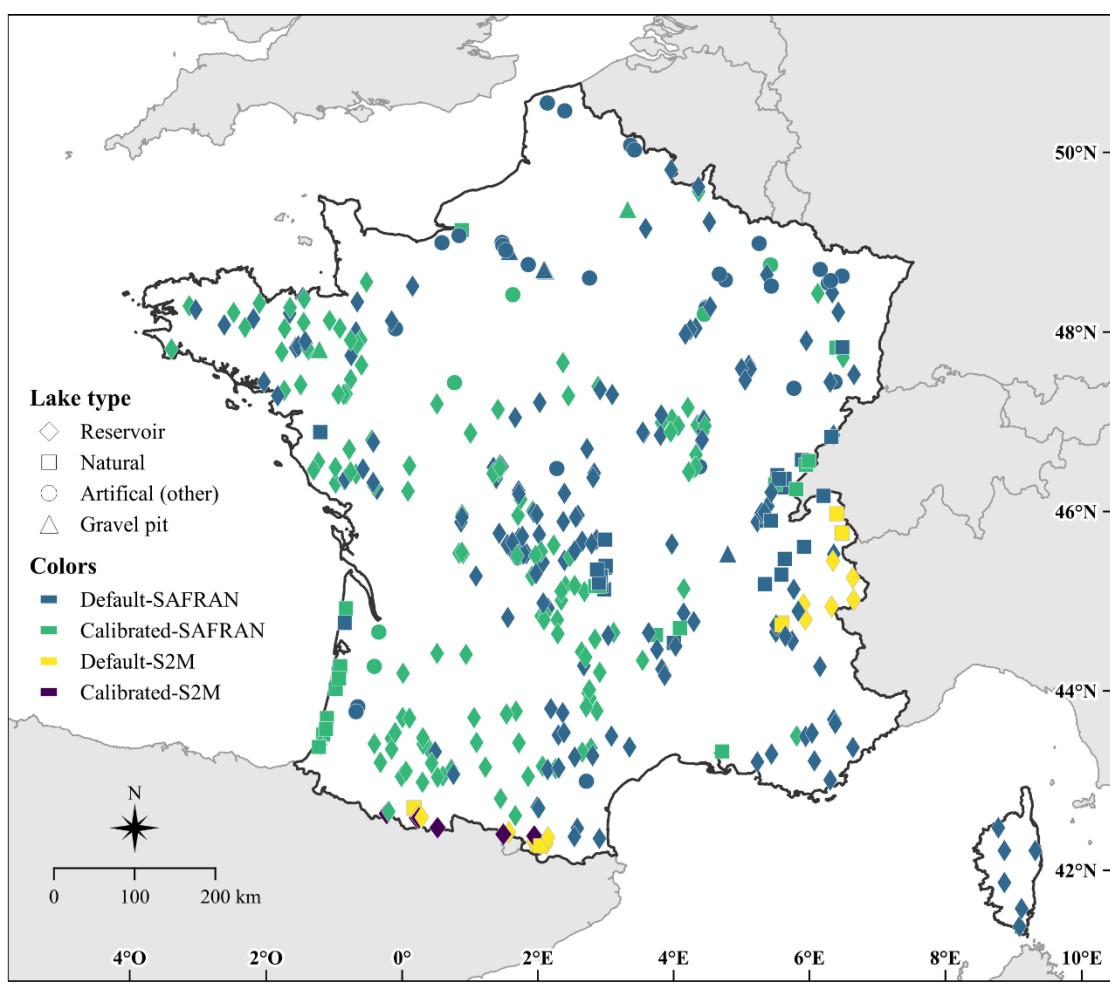


**Figure 1: Location and lake type of the 401 French lakes simulated with the OKPLM in "default" and "calibrated" modes, with SAFRAN and S2M meteorological data for the period 1959-2020. The "other" artificial lakes consist of ponds and quarry lakes.**

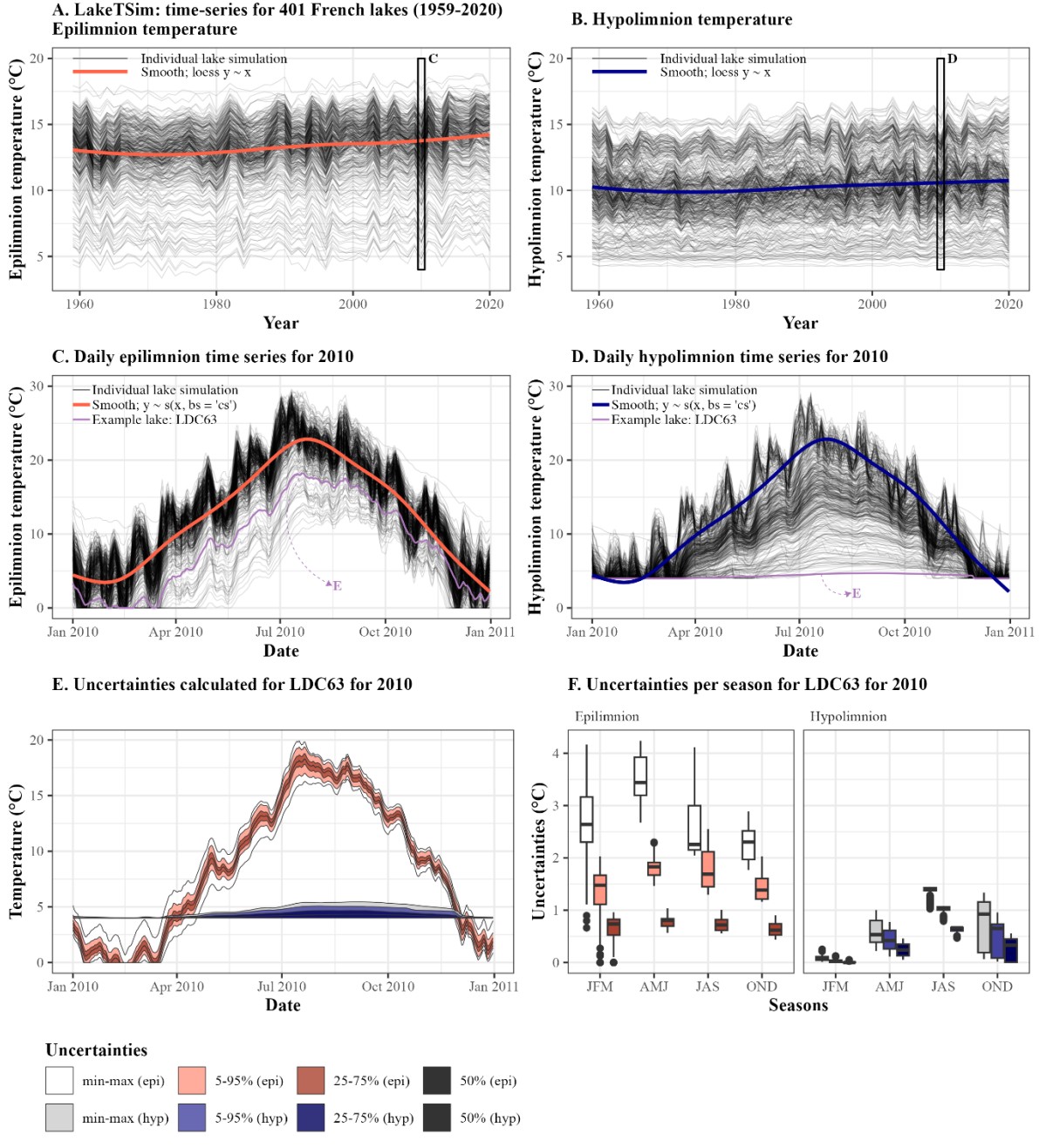

**Figure 2: Presentation of the LakeTSim data. (A) Epilimnion and (B) hypolimnion mean annual temperatures, with average trend across lakes shown with a smooth spline. (C) Daily epilimnion temperature per lake in the dataset for 2010, with smooth spline and the time series for one lake (LDC63) highlighted. (D) Daily hypolimnion temperature per lake in the dataset for 2010, with smooth spline and the time series for one lake (LDC63) highlighted. LDC63 is the code for Lake Chauvet, a natural lake (45.46 °N, 2.83 °E) located at 1167 m asl, with a surface area of 0.51 km$^2$, a volume of 17.41 10$^6$ m$^3$, and a maximum depth of 66.8 m. The simulation for LDC63 was conducted resorting to SAFRAN data and was run with the "calibrated" mode. (E) Uncertainties were calculated per lake and per day and are shown here daily for LDC63, in 2010, for both the epilimnion (epi) and the hypolimnion (hyp). (F) Uncertainties are shown here seasonally for LDC63, in 2010, for both the epilimnion (epi) and the hypolimnion (hyp). JFM corresponds to January-February-March, AMJ corresponds to April-May-June, JAS corresponds to July-August-September and OND corresponds to October-November-December.**

**Table 1: Characteristics of the lakes simulated with the OKPLM in "default" and "calibrated" modes with SAFRAN and S2M meteorological data for the period 1959-2020; *n* represents the number of lakes.**

| Variables | Minimal - Maximal range | | | |
|---|---|---|---|---|
| **Model parameters** | **Default** | | **Calibrated** | |
| **Meteorological data** | **SAFRAN** | **S2M** | **SAFRAN** | **S2M** |
| ***n*** | 210 | 21 | 164 | 6 |
| **Altitude (m)** | 1 - 1753 | 916 - 2213 | 0 - 2279.7 | 1577.5 - 2172.5 |
| **Latitude (°N)** | 41.47 - 50.87 | 42.55 - 46.21 | 42.88 - 49.87 | 42.65 - 42.86 |
| **Longitude (°E)** | -3.90 - 9.48 | 0.08 - 6.94 | -4.24 - 6.96 | -0.33 - 1.92 |
| **Maximal depth (m)** | 0.8 - 309.7 | 10.3 - 180 | 1.2 - 124 | 49 - 112 |
| **Surface area (km²)** | 0.08 - 577.12 | 0.11 - 6.52 | 0.29 - 57.57 | 0.45 - 1.21 |
| **Volume (m³)** | $5 \times 10^4 - 8.9 \times 10^{10}$ | $51.4 \times 10^4 - 33.32 \times 10^7$ | $12.9 \times 10^4 - 49.88 \times 10^7$ | $72.7 \times 10^5 - 68.6 \times 10^6$ |


**3.5.    Calibration, uncertainty and sensitivity analysis**
Calibration, uncertainty and sensitivity analyses of model parameters were carried out using the package "CUSPY"
(Calibration, Uncertainty analysis and Sensitivity analysis in PYthon), which is part of the software suite
"ALAMODE" (Danis, 2020) and acts as an interface to the package "pyemu" (White et al., 2016, 2020). In addition
to model parameters, sensitivity analysis was extended to encompass forcing parameters (*MAAT, at_factor,*
*sw_factor*) as they provide information about the degree of sensitivity exhibited by model parameters in response
to biases in the forcing data.
Parameter values were calibrated for lakes with available in situ data (temperature profiles and bathymetry).
Parameter values were calibrated using the Gauss-Levenberg-Marquardt algorithm and Tikhonov regularization
(White et al., 2020), and the squared sum of residuals as objective function. In addition to the calibrated parameter
values, the calibration process also provided posterior parameter uncertainty and composite scaled sensitivities.
Composite scaled sensitivities (*CSS*) indicate the quantity of information provided by each parameter and the
sensitivity of the model to them (Ely, 2006). The parameters with higher *CSS* values will have a greater impact on
the resulting simulation compared to those with low *CSS* values. To determine the *CSS* values for each parameter,
the Dimensionless Scaled Sensitivities (*DSS*) are used. *DSS* indicate how important an observation or how sensitive
a simulated equivalent of an observation is in relation to the estimation of a parameter. Further information on
these statistical measures is available in Hill (1998) and Poeter & Hill (1997). The dimensionless scaled sensitivity
for *i* and *j*, *i* being one of the observations and *j* being one of the parameters, is calculated as:
$\qquad DSS_{i,j} = \left[\frac{\partial y'_i}{\partial b_j}\right] b_j w_i^{1/2}$ \hfill (13)
where $y_i'$ is the simulated value associated with the *i*th observation, $b_j$ is the *j*th estimated parameter, $\frac{\partial y_i'}{\partial b_j}$ is the
sensitivity of the simulated value associated with the *i*th observation and $w_i$ is the weight of the *i*th observation
calculated based on the inverse of the variance-covariance matrix of the observation errors.
The *CSS* for parameter *j* is calculated from *DSS* as follows:
$$CSS_j = \left[ \frac{\sum_{i=1}^{ND}(DSS_{ij})^2|_b}{ND} \right]^{1/2}$$                                            (14)
where $ND$ is the number of observations and $\boldsymbol{b}$ is a vector of parameters values.
The uncertainty of the simulations (calibrated and default) was analyzed using Monte Carlo simulations. For each
lake, 100 Monte Carlo simulations were carried by randomly selecting the value of the model parameters. Two
parameters, $at\_factor$ and $sw\_factor$, multiplying the meteorological input, were added to account for possible
uncertainties in input data. For default simulations, the a priori distribution of the parameters was assumed to
follow a normal distribution with the average value and lower and upper bounds shown in Table 2. The ranges for
parameters $A$, $B$ and $C$ were estimated as four times the standard deviation of the residuals of the formulas used to
estimate them according to Prats & Danis (2019). The parameters $D$, $E$ and $\beta$, are expected to lie in the range 0-1
for mathematical and physical reasons. However, their respective values are highly interdependent and are difficult
to identify. Given their higher uncertainty, the full 0-1 range was explored. For $MAAT$, $at\_factor$ and $sw\_factor$,
reasonable ranges ($\pm10\%$) were chosen to account for meteorological data uncertainty (measurement error, errors
in regionalization, etc.). For calibrated simulations, the distribution of the parameters was obtained from the
calibration results.
In this study, the non-parametric Kendall's tau coefficient (significance level at 5%) was used to identify statistical
associations between uncertainty values and *CSS* in respect to lake geomorphological characteristics (maximal
depth, volume, surface area, latitude and altitude).









**Table 2: Characteristics of the a priori distributions of the model parameters. Parameters with a circumflex accent indicate parameter values estimated for a particular lake according to the regionalization formulas by Prats & Danis (2019).**

| Parameter | Average value | Lower bound | Upper bound |
|:---:|:---:|:---:|:---:|
| $A$ | $\hat{A}$ | $\hat{A} - 2 \cdot 0.74$ | $\hat{A} + 2 \cdot 0.74$ |
| $B$ | $\hat{B}$ | $\hat{B} - 2 \cdot 0.08$ | $\hat{B} + 2 \cdot 0.08$ |
| $C$ | $\hat{C}$ | $\hat{C} - 2 \cdot 0.004$ | $\hat{C} + 2 \cdot 0.004$ |
| $D$ | $\hat{D}$ | 0 | 1 |
| $E$ | $\hat{E}$ | 0 | 1 |
| $\alpha$ | $\hat{\alpha}$ | 0 | $\hat{\alpha} + 2 \cdot 0.08$ |
| $\beta$ | $\hat{\beta}$ | 0 | 1 |
| $MAAT$ | $M\hat{A}AT$ | $M\hat{A}AT - 2 \cdot 0.5$ | $M\hat{A}AT + 2 \cdot 0.5$ |
| $at\_factor$ | 1 | 0.9 | 1.1 |
| $sw\_factor$ | 1 | 0.9 | 1.1 |


## 4. Model performance
The performance of the OKPLM was assessed in Prats & Danis (2019) by comparing its performance to two other
often-applied models in lake studies, air2water (the 4-parameter version) and FLake. The air2water model is a
semi-empirical model used to calculate the epilimnion temperature of temperate lakes (Toffolon et al., 2014;
Piccolroaz et al., 2013). FLake is a one-dimensional (1D) hydrodynamic lake model for simulating temperature
vertical profiles and mixing conditions in lakes (Mironov, 2008). To assess their performances, the three models
were run between 1999 and 2016 over two sets of French lakes of different types (reservoirs, natural lakes, ponds,
quarry lakes and gravel pits): a group of five lakes with continuous profile measurements, and a group of 404 lakes
with less frequent temperature measurements. The performance assessment was limited to the period of 1999-2016
due to the availability of water temperature data (in situ and satellite) during that specific timeframe. The scarcity
of in situ water temperature measurements before 1999 applies to the entire set of lakes. However, it is important
to note that long-term in situ water temperature data is available for a few large lakes, which was used to assess
the performance of the three models (Prats & Danis, 2015). The OKPLM was run with the "default" parameter
values given by the parameterization in Prats & Danis (2019). The air2water parameter values were obtained as a
function of lake depth from the parametrization presented in Toffolon et al. (2014), based on data from 14 lakes
around the globe. In this case, the air2water model parameters were not calibrated due to the fact that the percentage
of missing data within the LakeSST dataset employed in Prats & Danis (2019) exceeded 97% for most lakes.
Beyond this threshold of 97% missing data, the performance of the calibrated 4-parameter version of the air2water
model was found to be unsatisfactory (Piccolroaz, 2016). However, when evaluating the model performance with
the set of five lakes with continuous data, air2water was run using parameter values calibrated for the individual
lakes available data. FLake does not have calibration parameters. Meteorological forcing (SAFRAN) consisted of
air temperature for the air2water model; solar radiation, vapor pressure, cloud cover and wind speed for FLake;
and air temperature and solar radiation for the OKPLM.
The OKPLM, air2water and FLake simulations were assessed through comparison to in situ measurements. For
epilimnion temperatures, the average discrepancies calculated between OKPLM simulations and observations
remained below 2 °C in most cases, in contrast to the air2water and Flake models. The performance comparison
between the OKPLM, air2water and FLake yielded respectively median RMSE's (Root Mean Square Error) of
1.7, 2.3 and 2.6 °C calculated between simulations and observations of epilimnion water temperature. Although
when using calibrated parameter values for air2water, median RMSE was below 1 °C in most cases. For
hypolimnion temperatures, the median RMSEs by lake type obtained with OKPLM simulations remained below
2 °C, except for gravel pits (RMSE = 2.7 °C) and reservoirs (RMSE = 2.3 °C), whereas FLake yielded a median
RMSE of 3.3 °C. For the epilimnion, the differences between the RMSE of lake types were not significant. In
terms of depth, discrepancies between epilimnion temperature simulations with the OKPLM and measurements
were highest for lakes with a depth > 10 m and for ponds around 1 m deep. The OKPLM simulations were also
evaluated seasonally, in particular during summer and winter. The model simulated temperatures well with a
median RMSE of 1.4 and 1.6 °C in summer and winter respectively.
## 5.   Uncertainty analysis
Overall, for both simulations with default and calibrated model parameters, uncertainty was higher for
hypolimnion temperature compared to epilimnion temperature especially in reservoirs (Figure 3). In default
simulations, the uncertainty of simulated epilimnion temperatures showed a clear and strong relation with lake
maximal depth (Figure 3, Table 3). On one hand, maximal depth had the highest Kendall's tau value of 0.64 ($p$-
value < 0.0001), indicating a strong positive correlation with uncertainty followed by volume with a Kendall's tau
of 0.59 ($p$-value < 0.0001). Uncertainty increased with maximal depth and volume in particular for lakes with
depths greater than 10 m and volumes greater than $10^6$ m$^3$ (Figure 3). Overall, lakes with higher maximal depths
have higher volumes and are located at greater altitudes (Figures A1-A2 in Appendix A). On the other hand,
moderate significant correlations were identified with surface area, altitude and latitude (Table 3). Lakes with
larger surface areas and higher altitudes tend to have higher uncertainties whereas lakes located at higher latitudes
tend to have lower uncertainties (Figure A3 in Appendix A). The latter can be linked to the fact that more shallow
lakes are located at higher latitudes (Figure A1 in Appendix A). For default simulations of hypolimnion
temperatures, uncertainty was maximal for lakes with depths around 10 m. Kendall's tau values revealed a
moderate significant correlation between hypolimnion temperature uncertainty and altitude (-0.45, $p$-value <
0.0001). The decrease in uncertainties with altitude can be related to the fact that lakes situated at very high
altitudes are mostly deep. Further, in the present dataset, lakes with higher maximal depths occur as altitude
increases (Figure A1-A2 in Appendix A).
After calibration, there was an important reduction in simulation uncertainty. For default simulations of epilimnion
temperature the median of the 90% confidence uncertainty range was 5.42 °C, while after calibration it was 1.85
°C. For hypolimnion temperature, the median of the 90% confidence uncertainty range of default simulations was
8.5 °C, while it was 2.32 °C after calibration. However, many reservoirs with depths greater than 8 m still had a

much greater uncertainty (uncertainty range > 4 °C) than the rest of lakes after calibration. Additionally, reservoirs (and a few natural lakes) above 100 m in altitude showed the highest uncertainties in the simulation of epilimnion temperature.

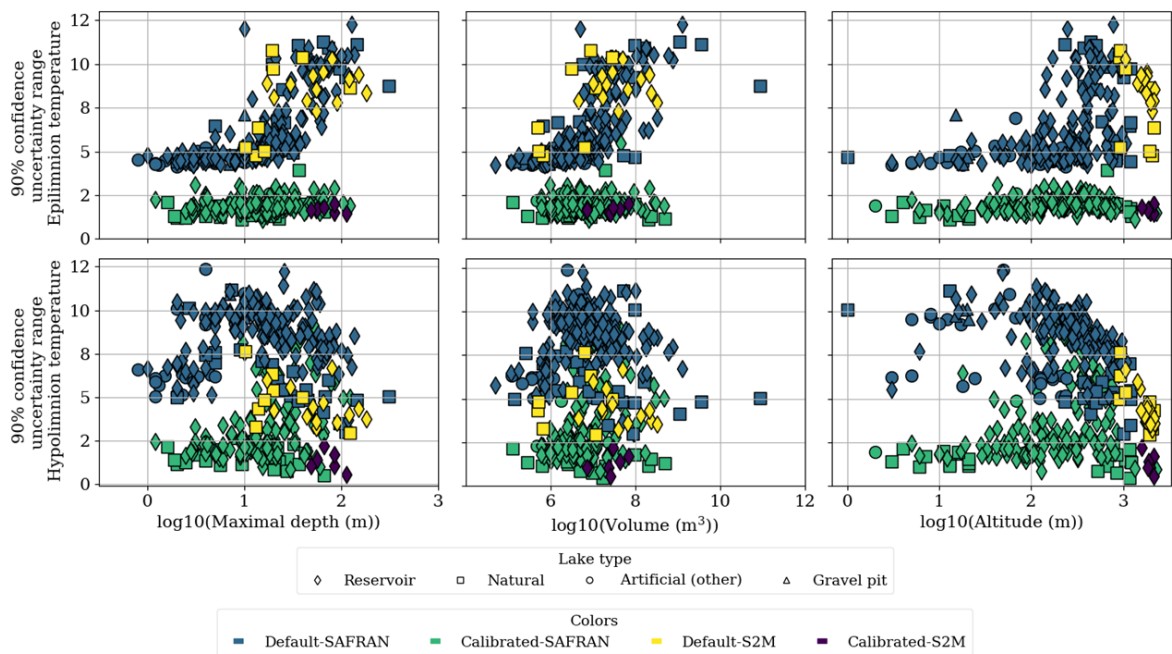

**Figure 3: Average 90% confidence uncertainty range for epilimnion (top panel) and hypolimnion (bottom panel) temperatures in calibrated ($n$ = 170) and default ($n$ = 231) simulations for the period 1959-2020. The "other" artificial lakes consist of ponds and quarry lakes.**

**Table 3: Kendall's tau coefficients and *p*-values of average 90% confidence uncertainty range for epilimnion and hypolimnion temperatures obtained from default simulations (1959-2020) in respect to lakes geomorphological characteristics. For each lake, "Epilimnion uncertainty" and "Hypolimnion uncertainty" are defined as the average 90% confidence uncertainty range calculated as the difference between the 95th and 5th percentiles of the daily simulated epilimnion and hypolimnion water temperatures. The significance levels are represented as follows: \*: 1.00e-02 < *p*-value ≤ 5.00e-02, \*\*: 1.00e-03 < *p*-value ≤ 1.00e-02, \*\*\*: 1.00e-04 < *p*-value ≤ 1.00e-03, \*\*\*\*: *p*-value ≤ 1.00e-04. Otherwise, correlations are not significant (*p*-value > 0.05).**

|  | Maximal depth (m) | Surface area (km$^2$) | Altitude (m) | Latitude (°N) | Volume (m$^3$) |
|---|---|---|---|---|---|
| **Epilimnion uncertainty** | 0.64\*\*\*\* | 0.31\*\*\*\* | 0.39\*\*\*\* | -0.40\*\*\*\* | 0.59\*\*\*\* |
| **Hypolimnion uncertainty** | -0.13\*\* | 0.05 | -0.45\*\*\*\* | 0.03 | -0.03 |

## 6. Sensitivity analysis

The parameter to which the model was most sensitive was the parameter *C* (Figure 4), which multiplies solar radiation in Eq. (1). The *CSS* for *C* were an order of magnitude greater than for the next parameters with highest *CSS*, the parameter *α* and *at_factor*, both influencing the effect of air temperature on simulated water temperature. Other parameters to which the model was somewhat sensitive were *E*, *B* and *β*. The model was quite insensitive to *sw_factor*, *MAAT* and *A*. The parameter *D*, with *CSS* several orders of magnitude smaller than the other parameters, was unidentifiable.

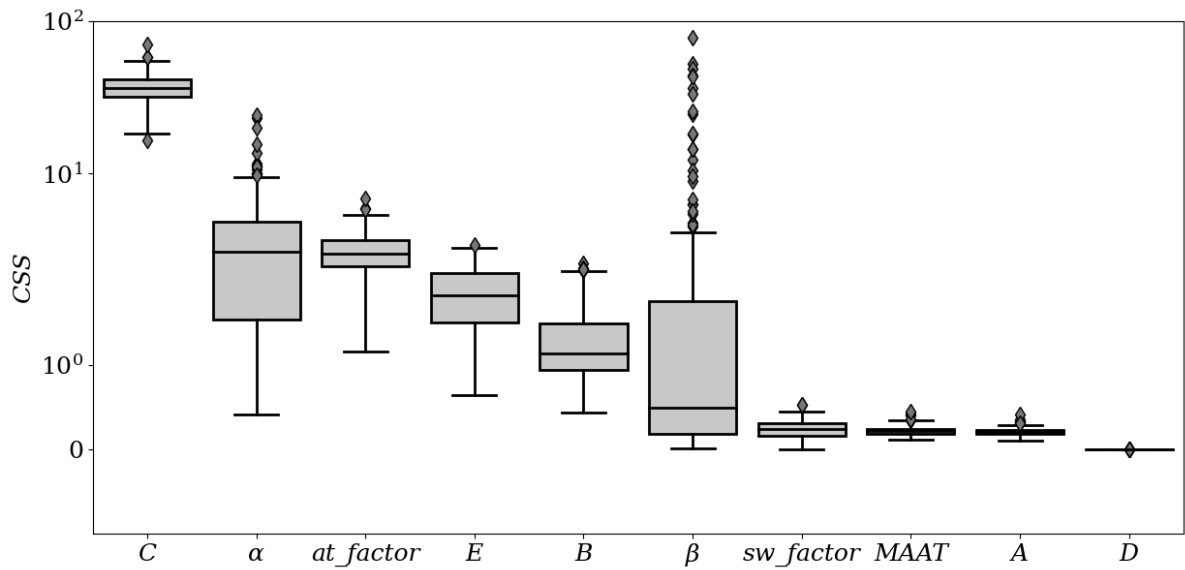

**Figure 4: Composite scaled sensitivities (*CSS*) for each parameter. The boxplots indicate the distribution of *CSS* between the simulations calibrated for different lakes. The y-axis is in logarithmic form.**

The model tended to be more sensitive to the parameter values in the case of reservoirs than in the case of natural lakes (Figure 5, Figures A4-A7 in Appendix A). Some parameters showed a dependency on lakes geomorphological characteristics. With the exception of a weak correlation with altitude (Kendall's tau = 0.18), there was no significant dependence between the parameter *C* and lakes geomorphological characteristics (Table 4, Figure A4 in Appendix A). The parameter α being parametrized as a function of lake volume, surface area and altitude reflects the thermal inertia of the lake. It showed a clear highly significant dependency primarily on lake depth (Kendall's tau = 0.47) followed by altitude (Kendall's tau = 0.4) and volume (Kendall's tau = 0.39) (Figure 5, Table 4). The increase of model sensitivity to the parameter α primarily with depth as well as altitude and volume propagated to the default simulations and explain the increased uncertainty with these same geomorphological characteristics in the default simulations. The parameter *at_factor*, was weakly but significantly correlated with all lakes geomorphological characteristics except for latitude with which no correlation was found (Figure 5, Table 4, Figures A4-A7 in Appendix A). *CSS* were mostly low for the parameter *β*, except for a few reservoirs and artificial lakes that scored very high *CSS* values. The sensitivity of *β* displayed a weak but significant correlation with lakes geomorphological characteristics, except for volume (Table 4).

Although the model in general was not very sensitive to the values of the parameters most directly related with hypolimnion temperatures (*D*, *E*, *β*), the quality of hypolimnion temperature was greatly improved through calibration. This would seem to indicate that the quality of simulated hypolimnion temperature was improved through the improvement of epilimnion temperature simulations.

**Table 4: Kendall's tau coefficients and *p*-values of *CSS* for model parameters values and drivers obtained from calibrated simulations (1959-2020) in respect to lakes geomorphological characteristics. The significance levels are represented as follows: \*: 1.00e-02 < *p*-value ≤ 5.00e-02, \*\*: 1.00e-03 < *p*-value ≤ 1.00e-02, \*\*\*: 1.00e-04 < *p*-value ≤ 1.00e-03, \*\*\*\*: *p*-value ≤ 1.00e-04. Otherwise, correlations are not significant (*p*-value > 0.05).**

| | Maximal depth (m) | Surface area (km$^2$) | Altitude (m) | Latitude (°N) | Volume (m$^3$) |
|---|---|---|---|---|---|
| $CSS_A$ | 0.02 | -0.1 | 0.14** | -0.08 | -0.07 |
| $CSS_B$ | 0.09 | -0.04 | 0.14** | -0.14** | 0.02 |
| $CSS_C$ | -0.04 | -0.09 | 0.18*** | -0.05 | -0.1 |
| $CSS_D$ | -0.12* | 0.02 | -0.14** | 0.06 | -0.1 |
| $CSS_E$ | -0.01 | -0.001 | 0.02 | 0.0003 | -0.03 |
| $CSS_\alpha$ | 0.47**** | 0.07 | 0.4**** | -0.23**** | 0.39**** |
| $CSS_\beta$ | 0.16** | -0.12* | 0.22**** | -0.19*** | 0.05 |
| $CSS_{at\_factor}$ | -0.25**** | -0.14** | -0.13* | 0.04 | -0.28**** |
| $CSS_{sw\_factor}$ | -0.22**** | -0.06 | -0.14** | 0.06 | -0.2**** |
| $CSS_{MAAT}$ | -0.09 | -0.13** | 0.13* | -0.02 | -0.15** |


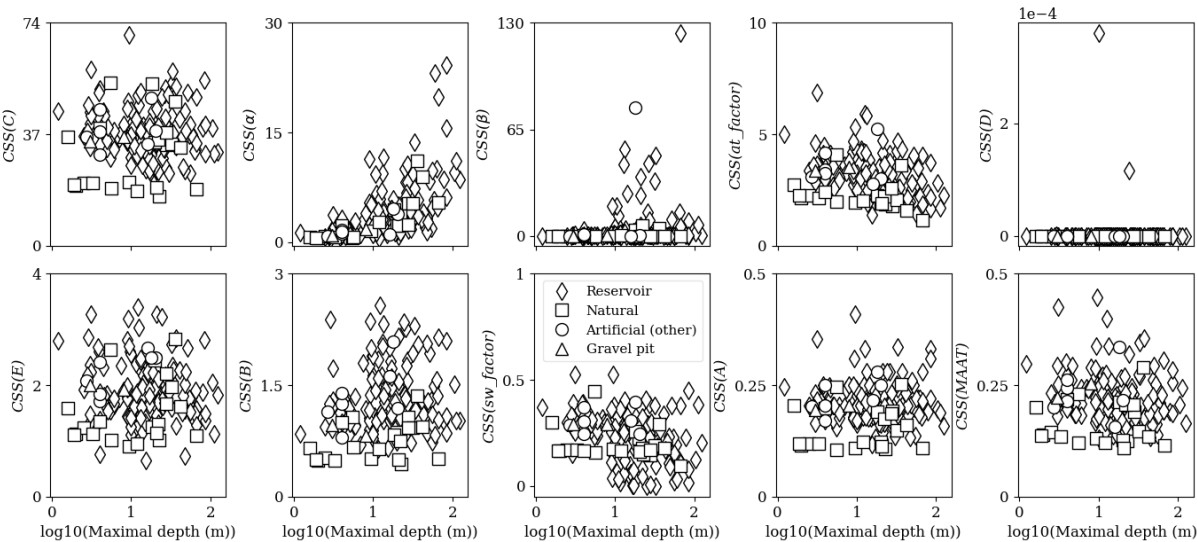


**Figure 5 : Composite scaled sensitivities (*CSS*) for each model parameter as a function of maximal depth. The "other" artificial lakes consist of ponds and quarry lakes.**

**7.    Discussion and implications**
Lakes are undeniably changing under climate change and long-term future projections show that the shifts in
ecosystem functioning will continue with aggravated alterations (Woolway & Merchant, 2019). In particular, given
the key role of lake water temperature in regulating ecosystem processes, its warming has become a response that
is crucial to monitor, explore and understand. Hence, the importance of developing or adopting approaches, such
as numerical models, that will provide long-term information about water temperature and allow us to understand
the thermal response of lakes to climate change.
Here we used a semi-empirical model, the OKPLM, to simulate six decades of epilimnion and hypolimnion water
temperatures in French lakes. In comparison to similar models, overall, the OKPLM provides acceptable
estimations of water temperatures, with better results for epilimnion temperatures. The values of the RMSEs
provided in Prats & Danis (2019) and obtained between OKPLM simulations and observations are comparable to
values found in studies applying complex hydrodynamic lake models (Read et al., 2014; Fang et al., 2012). When
using the default parameter values, the uncertainty associated with epilimnion temperature simulations was
significantly related to all geomorphological characteristics however, it was especially strongly correlated to lake
maximal depth. In contrast, the uncertainty in the hypolimnion simulations had a significant correlation solely with
altitude and maximal depth. The importance of this correlation was especially noteworthy in the case of reservoirs
located in low-altitude regions where uncertainties were the lowest. While the association between hypolimnion
uncertainty and maximal depth exhibited only a weak correlation, the instances of highest uncertainties were
predominantly found in reservoirs having maximal depths around 10 m. The correlations found between lakes
geomorphological characteristics and simulations uncertainties suggests that there might be systematic biases in
the definition of model parameters or in the forcing data. The calibration of model parameters significantly reduced
the uncertainties yet, for hypolimnion temperatures, they remained considerably high and increased with depth
especially in reservoirs.
The high levels of uncertainty found in reservoirs could be somewhat attributed to the lack of consideration of
water level fluctuations in the model. In contrast to other lakes (e.g., natural lakes, artificial lakes and gravel pits)
reservoirs experience significant variations in their water level, which influences the heat budget and hence their
thermal regime. Therefore, even under similar meteorological conditions lakes and reservoirs could have different
thermal behaviors (Nowlin et al., 2004). In reservoirs, the discharge depth is a driver of thermal structure. Deep
discharges could contribute to warmer bottom waters (Carr et al., 2020) whereas in some cases if the reservoir is
shallow or if the discharge depth is not deep, it could demonstrate lake-like thermal behavior. This does not
necessarily mean that, in this case, the entire functioning of the reservoir resembles one of a natural lake; there are
still differences to consider (Detmer et al., 2022).
The application of the OKPLM should be made with caution given its performance and depending on the objective
of the study. The model does not take into account a complete set of meteorological forcing (e.g., with cloud cover,
relative humidity and wind speed and direction) or other variables (e.g., inflow and outflow rates or water level
fluctuations, inflow discharge depth and inflow temperature) that could influence the thermal structure of the
ecosystem (Yang et al., 2020; Carr et al., 2020). Furthermore, the OKPLM was parametrized for a specific set of
lakes with particular geomorphological characteristics. Thus, it would be advisable to apply the model over lakes
with similar characteristics. If the aim is to conduct a long-term regional or global study for studying general
patterns of climate change impacts over a large number of study sites, the utilization of semi-empirical models
such as the OKPLM is the most suitable choice. Although complex, deterministic or process-based models provide
a more accurate representation of thermal conditions, applying these models over several study sites and for long
periods is usually hindered by the scarcity of the required input data. The increased complexity of these models
(with reference to an increased number of model parameters) is beneficial for representing additional ecosystem
processes. Yet the greater number of model parameters, increases the sensitivity of models and demands more
calibration efforts (Lindenschmidt, 2006). Furthermore, a reduction in model errors is sometimes associated with
an increased complexity in model structure; however, this is not always consistent since a complex model does
not necessarily provide better estimations and thus lower errors than a simple model (Snowling and Kramer, 2001).
Our goal in publishing the present dataset is to expand knowledge about the water temperature of French lakes and
provide data, with enough details and reliability, that it could be implemented in different studies where water
temperature is implicated for understanding specific processes or interactions, in particular under climate change.
Hence the significance of the present findings. The present study, making use of a semi-empirical model to provide
long-term data about water temperature, was necessary for several reasons. Equipping a large number of lakes
with thermal sensors is challenging and labor-intensive, it comes with a high financial cost that is often not
available. Consequently, historical and even current water temperature datasets are often scarce, which can be
problematic for studying the impact of climate change, as it requires high frequency data over a long duration of
time for accurate analysis. In general, the higher the sampling frequency and duration, the better the data is suited
to estimate or analyze specific processes or warming trends. The sampling frequency and length of a dataset have
been shown to play a role in determining the accuracy of estimating warming trends where time series longer than
30 years seem to be the most appropriate (Gray et al., 2018). Although, the duration and frequency of a dataset
have a major role in reflecting accurate representations, their influence is scarcely addressed when it comes to
climate change studies related to warming trends in water temperature.
This dataset will be useful for climate change studies; it could be used to develop and analyze several temperature
indicators (e.g., annual or seasonal maximal and minimal temperature values, temperature exceeding certain
thresholds with biological implications, etc.). Further, mixing and stratification dynamics are important to
characterize as they drive lake biogeochemistry. Among other processes, they influence the distribution of
nutrients, primary productivity and the composition of phytoplankton and zooplankton communities along the
water column (Judd et al., 2005). With the LakeTSim dataset, it is possible to classify the mixing regime of lakes
and investigate possible triggers of regime shifts.
**8. Data usage**
The LakeTSim dataset comprises water temperature simulations for natural lakes ($n = 54$), reservoirs ($n = 302$),
gravel pits ($n = 7$), and other artificial lakes (e.g., ponds and quarry lakes, $n = 38$). The simulations are for both the
epi- and hypolimnion. Lakes that are fully mixed throughout the year (typically, shallower lakes) have the same
temperature value for both layers. More generally, the delta of temperature can be used to calculate mixing regimes
(Sharaf et al., in prep.).
The lakes in the dataset were selected because they are monitored as part of the European Water Framework
Directive (Directive 2000/60/EC). The majority of the 401 lakes are non-natural and some were only created after
1959 (i.e., the start of our simulations). We compiled the initial temporal gap filling related to the initial filling
years for 282 of these 347 non-natural lakes (269 reservoirs and 13 artificial lakes, Figure A8 in Appendix A) in
Table S1 (see Supplement) to be used as a companion dataset to LakeTSim. The filling years were sourced from
https://www.barrages-cfbr.eu for 179 of the lakes and from the PLAN_DEAU database for 103 of the lakes; the
information was not available for 33 reservoirs, 7 gravel pits and 25 other artificial lakes of the LakeTSim dataset.
The median filling date was 1962 and 67% of the lakes with known filling dates were filled by 1980. While the
complete simulations ranging from 1959 to 2020 can also be used as theoretical lake temperature for comparison

across similar periods, we recommend that users of LakeTSim data for reservoir and artificial lake simulations consider the initial filling dates provided in Table S1 to filter out years from the simulations during which lakes were not filled yet.

Additionally, users should be aware that some reservoirs might be drained completely at certain intervals (e.g., every 10 years) for maintenance and inspection purposes, and that this is not reflected in our dataset. Finally, as mentioned in the discussion, some of the lakes in the dataset experience artificial (e.g., in reservoirs) or natural (e.g., in some smaller ponds) water level fluctuations, and potential intermittent dry-periods lasting weeks or months; none of these hydrological processes are accounted for in the simulations.

### 9. Code availability

The respective codes for the "CUSPY" (Prats-Rodríguez and Danis, 2023a) and "OKPLM" (Prats-Rodríguez and Danis, 2023b) packages, which can be used to conduct sensitivity and uncertainty analysis and to run the OKP Lake Model, are available at https://github.com/inrae/ALAMODE-cuspy and https://github.com/inrae/ALAMODE-okp as well as ZENODO.

### 10. Data availability

The LakeTSim dataset (Sharaf et al., 2023) for epilimnion and hypolimnion water temperature simulations and supporting information are available at doi:10.57745/OF9WXR . The file "00_Data_description.txt" contains a description of the dataset. The geographical (longitude and latitude) and morphological (surface area, volume and maximal depth) data for the 401 lakes are presented in the file "01_Lake_data.txt" in addition to the name, type, altitude and the identification code for each lake. The data are located in two main folders: "02_Temperature_data" containing daily epilimnion (tepi) and hypolimnion (thyp) temperatures simulated with the OKPLM and "03_Uncertainty_data" containing daily tepi and thyp uncertainties. In each folder, the data for temperature simulations and their uncertainties are presented in text files available in the folders "00_LakeTSim_SAFRAN_OKPdefault_data", "01_LakeTSim_SAFRAN_OKPcalibrated_data", "02_LakeTSim_S2M_OKPdefault_data" and "03_LakeTSim_S2M_OKPcalibrated_data". The name of each file within these folders includes the identification code of the lake. From 2024, the data will be visible from a dashboard. The link to the dashboard will be accessible from data.ecla.inrae.fr.

### 11. Conclusions

We present the LakeTSim dataset and the semi-empirical OKP Lake Model for simulating water temperature in lakes. We applied the model over a set of 401 French lakes for the period 1959-2020 to derive daily simulations of epilimnion and hypolimnion water temperatures, here referred to as the LakeTSim dataset. Previous efforts to assess the model's performance show an overall acceptable representation of epilimnion and hypolimnion temperatures when compared to in situ measurements. The uncertainty analysis of simulations demonstrates that higher uncertainties are found for, by order of relative importance: (1) default simulations, (2) hypolimnion compared to epilimnion temperatures and, (3) deep lakes, in particular reservoirs (maximal depth greater than 10 m for epilimnion temperature and around 10 m for hypolimnion temperature simulated with default model parameters). Although the calibration significantly decreases the uncertainties related to both the epilimnion and hypolimnion, in some cases they are still considerable in the hypolimnion. Based on these results and if enough observation data are available, optimally we recommend the use of the OKPLM for shallow (maximal depth < 8

m) lakes with calibrated model parameters. However, if applied in its default or even calibrated configuration over deep lakes, one should be aware of the presented limitations and address them in the analysis. The LakeTSim dataset is valuable for assessing the impact of climate change on lakes thermal functioning, which is often hindered by the lack of water temperature observations. The present dataset will provide new insights about the thermal behavior of French lakes, which can provide useful context for stakeholders as they design management strategies in a context of climate change.

 **12. Appendix A**

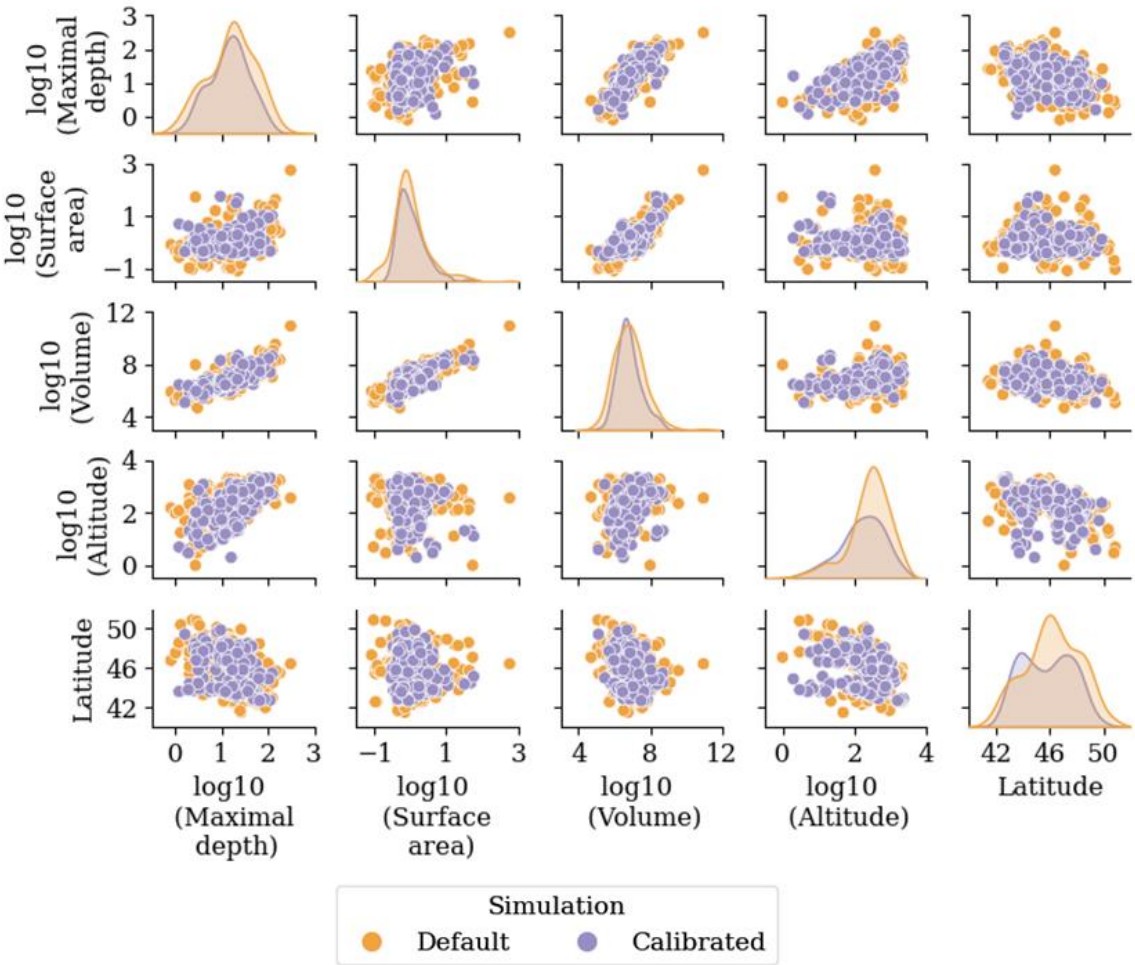

**Figure A1: Scatter plots of lakes (*n* = 401) geomorphological characteristics: Maximal depth (m), Surface area (km²), volume (m³), altitude (m) and latitude (°N).**

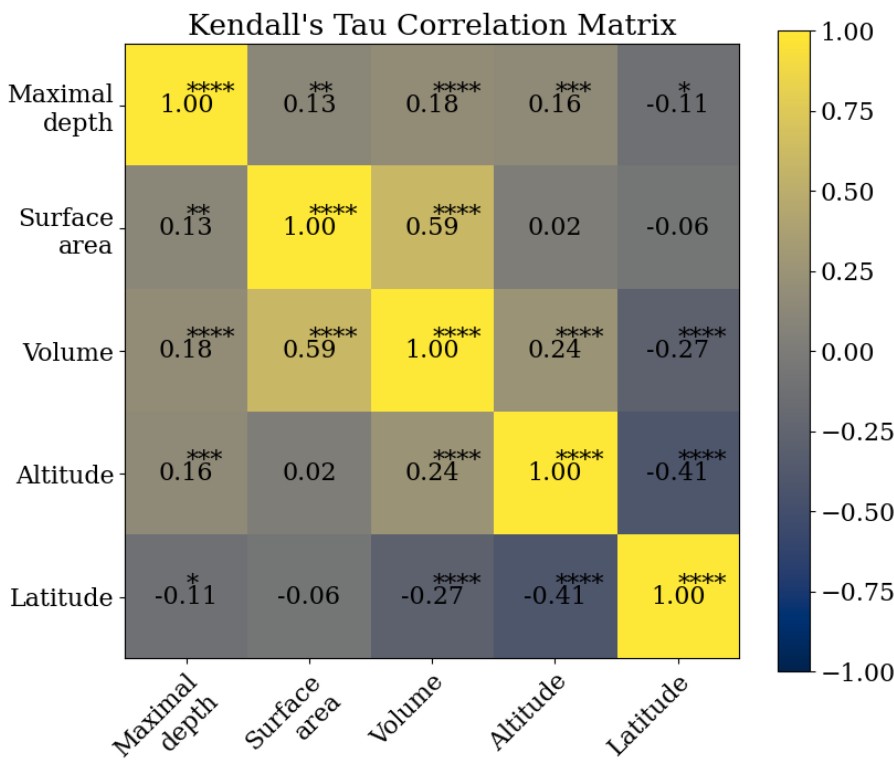

**Figure A2: Kendall's tau correlation matrix of the geomorphological characteristics of lakes simulated in "default" mode ($n = 231$): Maximal depth (m), Surface area (km$^2$), volume (m$^3$), altitude (m) and latitude (°N). The significance levels are represented as follows: \*: 1.00e-02 < *p*-value ≤ 5.00e-02, \*\*: 1.00e-03 < *p*-value ≤ 1.00e-02, \*\*\*: 1.00e-04 < *p*-value ≤ 1.00e-03, \*\*\*\*: *p*-value ≤ 1.00e-04. Otherwise, correlations are not significant (*p*-value > 0.05).**

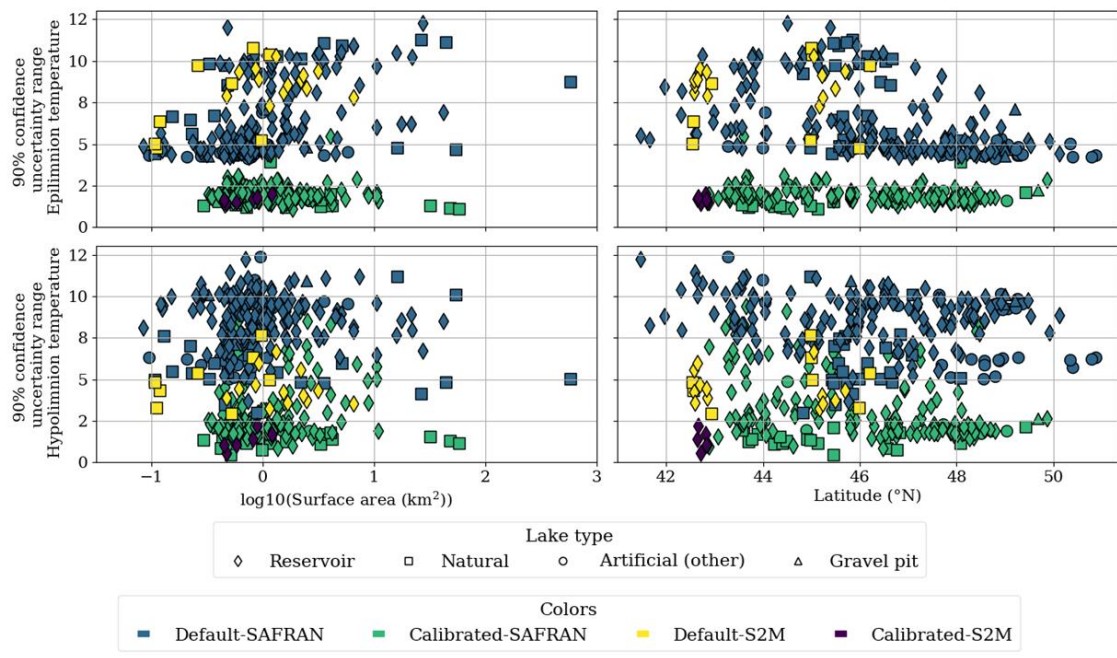

**Figure A3: Average 90% confidence uncertainty range for epilimnion (top panel) and hypolimnion (bottom panel) temperatures in calibrated ($n = 170$) and default ($n = 231$) simulations for the period 1959-2020 as a function of surface area (km$^2$) and latitude (°N). The "other" artificial lakes consist of ponds and quarry lakes.**

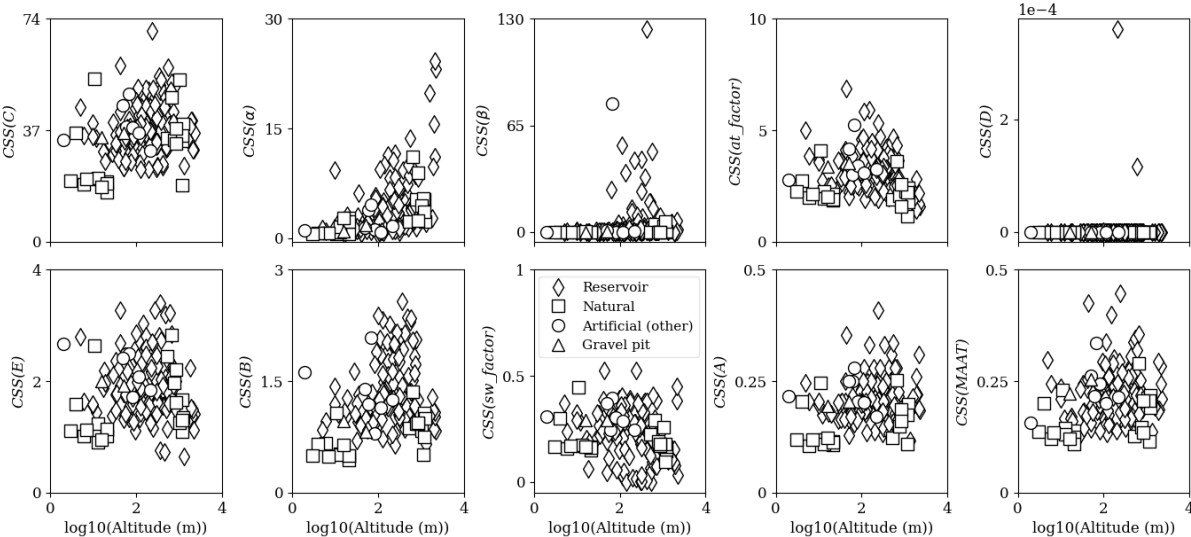

563

**Figure A4: Composite scaled sensitivities (*CSS*) for each model parameter as a function of altitude. The "other" artificial lakes consist of ponds and quarry lakes.**

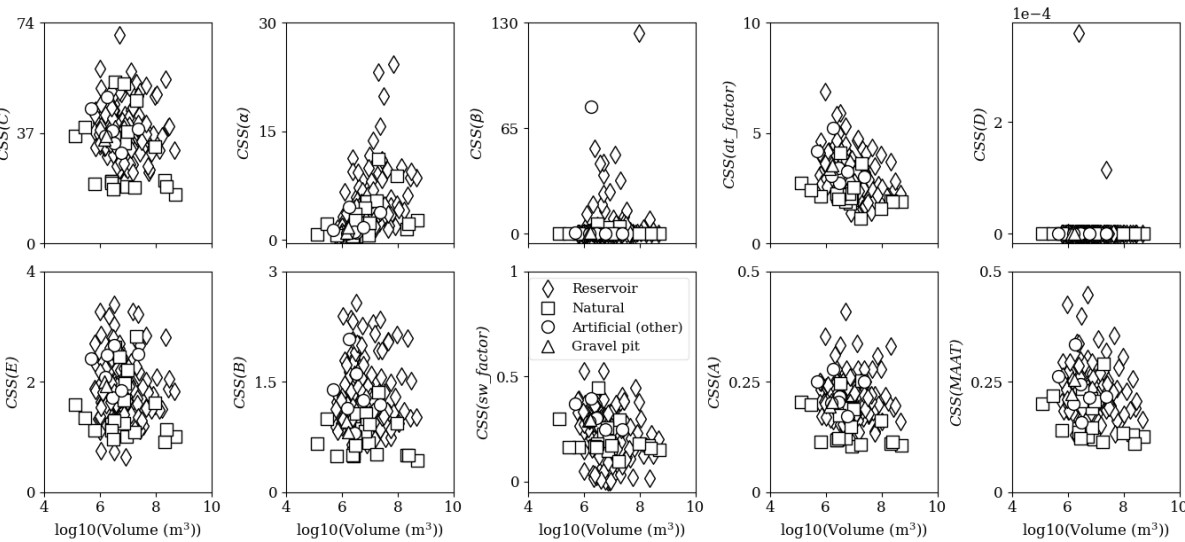

**Figure A5: Composite scaled sensitivities (*CSS*) for each model parameter as a function of volume. The "other" artificial lakes consist of ponds and quarry lakes.**

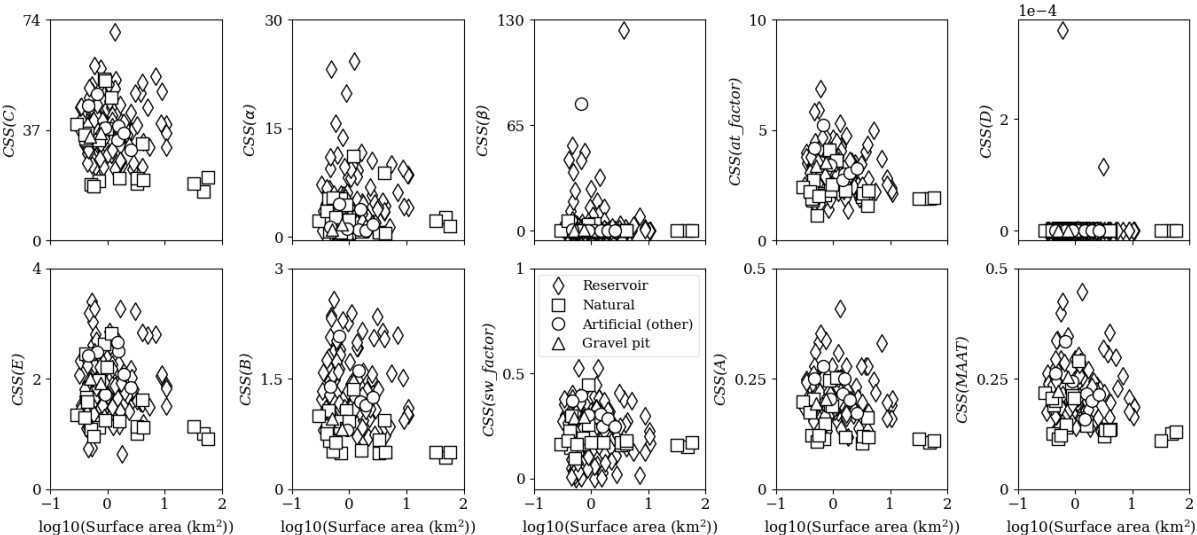

**Figure A6: Composite scaled sensitivities (*CSS*) for each model parameter as a function of surface area. The "other" artificial lakes consist of ponds and quarry lakes.**

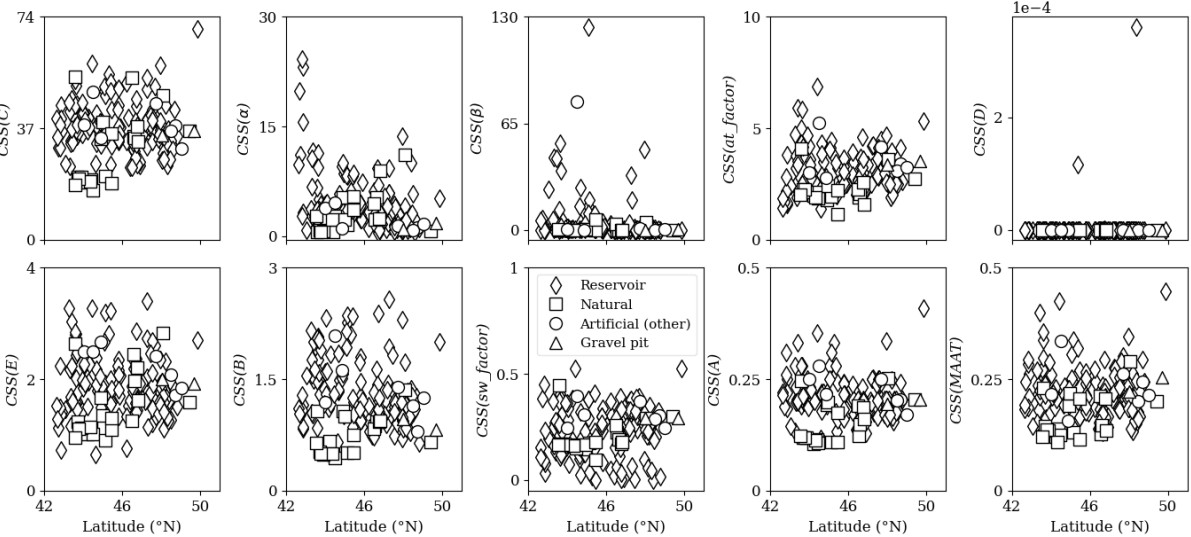

**Figure A7: Composite scaled sensitivities (*CSS*) for each model parameter as a function of latitude. The "other" artificial lakes consist of ponds and quarry lakes.**

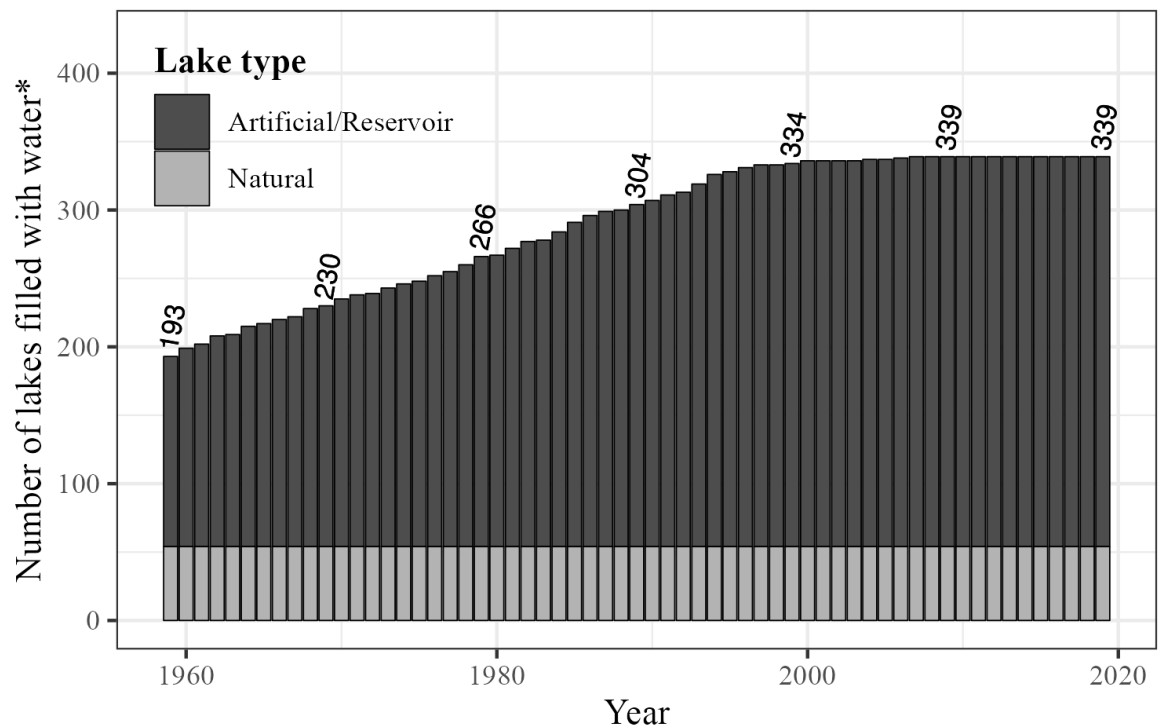

* 7 gravel pits, 33 reservoirs & 25 other artificial lakes with no information on filling year.

**Figure A8: Distribution of initial filling years for lakes (e.g., reservoirs, gravel pits, ponds and quarry lakes) of the LakeTSim dataset.**

### 13. Author contributions

NS wrote the original manuscript with input from JP and PAD. NS, JP and PAD discussed the results. JP developed and carried out the implementation of the OKP Lake Model and the uncertainties computation in ALAMODE. JP and NS performed the simulations and provided uncertainty analysis results with SAFRAN and S2M data respectively. JP and NS implemented respectively the integration of SAFRAN and S2M data in ALAMODE. NS prepared the LakeTSim dataset. JP and NS provided the uncertainty and sensitivity analysis. PAD designed, contributed and supervised the implementation of S2M data in ALAMODE for forcing the OKPLM when simulating high altitude lakes. PAD supervised the findings of this work. RB proposed and contributed to the integration of the data consisting of initial filling dates of reservoirs and other artificial lakes in the manuscript. NR and TT supervised and contributed to the implementation of simulation results in the database. NR processed S2M data and compiled the data for initial filling years of reservoirs and other artificial lakes. NS, RB, and NR prepared the figures. NR and TT prepared the doi for the LakeTSim dataset. TP conducted the fieldwork for the monitoring, acquisition and verification of in situ temperature data. All authors reviewed, edited and approved the final paper.

### 14. Competing interests

The authors declare that they have no conflict of interest.

**15. Acknowledgments**
The authors thank Météo-France for providing SAFRAN and S2M meteorological data, Matthieu Vernay for his
feedback on the utilization of S2M data and the "Réseau Lacs Sentinelles" for providing bathymetry data for
mountain lakes.
**16. Financial support**
The authors were supported by OFB (Office Français de la Biodiversité), SEGULA Technologies, INRAE (Institut
National de Recherche pour l'Agriculture, l'Alimentation et l'environnement) and Pôle R&D ECLA
(ECosystèmes LAcustres).

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
