# Peer review of "A long-term dataset of simulated epilimnion and hypolimnion temperatures in 401 French lakes (1959-2020)"

_Earth System Science Data, 2022_

## Referee Comment (RC1)

A long-term dataset of simulated epilimnion and hypolimnion temperatures in 401 lakes (1959-2020)

The manuscript presents a dataset of simulated lake water temperature of 401 French lakes using a modelling approach. The modelling approach is well explained as well as the choices of how to calibrate (or not) the model parameters. I appreciate the fact that authors discuss the limitations of the dataset as well as the suitable applications, making clear for which purposes this dataset would be limiting.

I think that the presented dataset is relevant to leverage lake research in the context of climate change. The manuscript it is overall well written but I suggest some modifications that I list below.

In the following paragraph, I collect the minor issues from my review.

**Minor issues:**

- L58 – "with the scarcity of….". I suggest rephrasing this sentence.
- L61 – "…epilimnion and hypolimnion temperature", I suggest adding "water" before temperatures.
- L62 – "We also….". I suggest rephrasing for consistency with the rest of the abstract.
- L71 – I suggest substituting "by" with "and".
- L89 – I suggest changing the beginning of the paragraph which sounds not very fluent.
- L93 – "rendering it challenging", please rephrase.
- L94 – "which is" should be removed.
- L105-107. I suggest to rephrase this sentence for clarity.
- L155 – please add reference at the end of the sentence reporting the ALAMODE model.
- Paragraph from L 188. I suggest adding the reason why sometimes SAFRAN has been used and some other times S2M has been used already in this paragraph otherwise the reader remains with this doubt for too long.
- L280 – I suggest adding a reference at the end of the first sentence of the paragraph.

---

## Author Response (AR1)

**Response letter**

**Preprint title (ESSD-2022-457):** A long-term dataset of simulated epilimnion and hypolimnion temperatures in 401 French lakes (1959-2020).

**Authors:** Najwa Sharaf, Jordi Prats, Nathalie Reynaud, Thierry Tormos, Rosalie Bruel, Tiphaine Peroux, Pierre-Alain Danis.

Reviewers comments are in _italic_ and authors comments are in **bold**.

**➕ Reviewer 1:**
_A long-term dataset of simulated epilimnion and hypolimnion temperatures in 401 lakes (1959-2020)_

_The manuscript presents a dataset of simulated lake water temperature of 401 French lakes using a modelling approach. The modelling approach is well explained as well as the choices of how to calibrate (or not) the model parameters. I appreciate the fact that authors discuss the limitations of the dataset as well as the suitable applications, making clear for which purposes this dataset would be limiting._

_I think that the presented dataset is relevant to leverage lake research in the context of climate change. The manuscript it is overall well written but I suggest some modifications that I list below._

**On behalf of my co-authors, I would like to thank the reviewer for taking the time to review our manuscript as well as for their comments and constructive feedback. All of their suggestions were taken into account and the manuscript was revised accordingly. Below we address each issue and provide in details the corresponding modifications made in the manuscript.**

_In the following paragraph, I collect the minor issues from my review._

_Minor issues:_

- _L58 – "with the scarcity of….". I suggest rephrasing this sentence._
  **Reformulated as follows (lines 58-60): "We combined numerical modelling and satellite thermal data to create a regional dataset (LakeTSim: Lake Temperature Simulations) of long-term water temperatures for 401 French lakes in order to tackle the scarcity of in situ water temperature."**

- _L61 – "…epilimnion and hypolimnion temperature", I suggest adding "water" before temperatures._
  **Done on line 60.**

- _L62 – "We also….". I suggest rephrasing for consistency with the rest of the abstract._
  **Reformulated as follows (lines 62-65): "Here, we describe the model and its performance. Additionally, we present the uncertainty analysis of simulations with default parameter values (parametrized as a function of lake characteristics) and calibrated parameter values, along with the analysis of the sensitivity of the model to parameter values and biases in the input data."**

- _L71 – I suggest substituting "by" with "and"._
  **Done on line 70.**

[Figure]

- *L89 – I suggest changing the beginning of the paragraph which sounds not very fluent.*
  **Reformulated as follows (lines 88-89): It is thus crucial to closely evaluate water temperature trajectories over the entire water column in space and time when assessing the impact of climate change on lake ecosystems.**

- *L93 – "rendering it challenging", please rephrase.*
  **Reformulated as follows (lines 89-94): "However, the lack of data coverage, both spatially and temporally, makes it difficult to accurately characterize lakes thermal response to climate change and to identify warming trends (Gray et al., 2018). Indeed, long-term datasets of in situ temperatures are usually scarce and mostly limited to large lakes (Layden et al., 2015). Moreover, sampling frequency and temporal coverage of in situ water temperature varies greatly from one lake to the next, from a few years (Sharma et al., 2015) up to decades (Piccolroaz et al., 2020; Rimet et al., 2020)."**

- *L94 – "which is" should be removed.*
  **Done on line 95.**

- *L105-107. I suggest to rephrase this sentence for clarity.*
  **Reformulated as follows (lines 105-107): "Long-term 105 simulations across a considerable number of lakes are made possible with this type of models, enabling the 106 detection of trends in time series data that are not achievable with shorter datasets (Gray et al., 2018)."**

- *L155 – please add reference at the end of the sentence reporting the ALAMODE model.*
  **We added an additional sub-section (3.1. The software suite ALAMODE) to section 3 "Data and methodology" in which we provide more details about ALAMODE and its utilization. (Danis, 2020) was used to reference ALAMODE.**

- *Paragraph from L 188. I suggest adding the reason why sometimes SAFRAN has been used and some other times S2M has been used already in this paragraph otherwise the reader remains with this doubt for too long.*
  **The reason for using S2M data is explained in the sentence "S2M data are more representative of mountainous meteorological conditions than SAFRAN data and were thus used for simulating the water temperature in lakes situated at altitudes higher than 900 m." previously located at lines 194-195. We moved it to lines 247-249 and reformulated as follows: "S2M data are more representative of mountainous meteorological conditions than SAFRAN data and were thus used, when possible, for simulating the water temperature in lakes situated at altitudes higher than 900 m."**

  **We also added additional information on the utilization of S2M data in section 3.3: Input data (lines 218-228).**

- *L280 – I suggest adding a reference at the end of the first sentence of the paragraph.*
  **We added the following reference** (Woolway and Merchant, 2019) **on line 387.**

[Figure]

[Figure]

[Figure]

✦ **Reviewer 2:**

*In this study, Sharaf and co-authors present a regional long-term water temperature dataset (LakeTSim: Lake Temperature Simulations) for 401 French lakes by combining numerical modelling and satellite thermal data. The dataset consists of daily epilimnion and hypolimnion temperatures. Simulations have been carried out using the semi-empirical OKPLM model. The authors also describe the model and its performance (including uncertainty and sensitivity analysis). The manuscript is clear and well written. Concise and nice to read, but probably too concise in some parts (see below). I have two main comments relative to sections 4 and 6, which are reported below:*

**On behalf of my co-authors, I would like to thank the reviewer for taking the time to review our manuscript and for their feedback. We have carefully considered the comments and suggestions provided by the reviewer and have addressed them in the following sections.**

*Section 4: Please explain what do you mean by "default parameter values" in the case of air2water and Flake. As for the first model, I do not think that default parameters are available.*
**The air2water model exists in two versions. The first one proposed by Piccolroaz et al. (2013) in which the model holds 8 parameters (p1, p2, p3, p4, p5, p6 p7 and p8). The second version proposed by Toffolon et al. (2014) and Piccolroaz (2016) in which the model is simplified to 4 parameters (a1, a2, a3 and a4 where a1 corresponds to p3 in that paper, a2 to p4, a3 to p4-p5 and a4 to p6). Prats & Danis (2019) used the latter (the 4-parameter version of air2water) where the OKP model was presented and its performance was assessed and compared to that of the air2water model. The air2water model parameters in Prats & Danis (2019) were applied with their default values, meaning that the values of the parameters (a1, a2, a3 and a4) used in Prats & Danis (2019) were obtained from the parametrization proposed in Toffolon et al. (2014) over a set of 14 lakes with different morphological characteristics (equations 15, 16, 17 & 18 in Prats & Danis 2019).**

**We reformulated and added some missing information about the calibration of air2water parameters for lakes with continuous profile measurements as follows (lines 310-313): "The air2water parameter values were obtained as a function of lake depth from the parametrization presented in Toffolon et al. (2014). When evaluating the model performance with the set of five lakes with continuous data, air2water was also run using parameter values calibrated for the individual lakes available data."**

**Regarding FLake, it does not have calibration parameters. We added this on line 313.**
**Thus, we removed the expression "default parameters values".**

*Also, it is unclear if the comparison discussed at lines 233-244 is against OKPLM run with default parameters (to be fair) or not.*
**To assess their performances, the three models were run between 1999 and 2016 over two sets of French lakes of different types (reservoirs, natural lakes, gravel pits and other artificial lakes including ponds and quarry lakes): a group of five lakes with continuous profile measurements, and a group of 404 lakes with less frequent temperature measurements. The OKPLM was run with the "default" parameter values given by the parameterization in Prats & Danis (2019). The air2water parameter values were obtained as a function of lake depth from the parametrization proposed in Toffolon et al. (2014) and were calibrated for five lakes for which continuous temperature profile measurements were available.**

**The comparison discussed at lines 316-320 shows the RMSE's calculated between observations and simulations respectively from the OKPLM, air2water and FLake. In this paragraph we show that, overall, the discrepancies obtained with the OKPKLM are lower, thus it is not necessarily against the OKPLM.**

*At line 229 the authors refer to 409 lakes, but in Fig. 1 and relative text to 401 lakes. Why there are 8 lakes more in the first case?*

The reference to the usage of 409 lakes in Prats & Danis (2019) was a mistake and has been removed.

The text has been modified to clarify this as follows (lines 302-305): "To assess their performances, the three models were run between 1999 and 2016 over two sets of French lakes of different types (reservoirs, natural lakes, ponds, quarry lakes and gravel pits): a group of five lakes with continuous profile measurements, and a group of 404 lakes with less frequent temperature measurements."

Although we use the same databases as Prats & Danis (2019) in this study, we do not use the same dataset. Prats & Danis (2019) made a direct extraction of data from the PLAN_DEAU database. In this work, we use the data integrated in the base TMOD (part of ALAMODE) from the PLAN_DEAU database a few years later. Differences in the queries used or modifications in the data base may be responsible for the difference in the number of simulated lakes.

*Was the forcing used to run OKPLM, air2water and Flake the same? At line 231 I understand that air2water and Flake were run only with SAFRAN.*
The same meteorological forcing, in this case SAFRAN meteorological data, was used when running the OKPLM, air2water and FLake models in Prats & Danis (2019). We reformulated and clarified this as follows (lines 313-315): "Meteorological forcing (SAFRAN) consisted of air temperature for the air2water model; solar radiation, vapor pressure, cloud cover and wind speed for FLake; and air temperature and solar radiation for the OKPLM."

*Why the comparison with air2water and Flake was not done considering the period 1959-2020?*
At the time of the preparation of the study by Prats & Danis (2019), where this comparison was made, the availability of water temperature data (satellite and in situ) was limited mostly to 1999-2016. In the present datapaper, we do not aim to compare the performance of the OKPLM to other models such as air2water and FLake, which was already made in Prats & Danis (2019) and is described in this manuscript. Also, the availability of water temperature measurements before 1999 is really scarce. Prats & Danis (2015) used monthly data from the IS OLA (https://si-ola.inrae.fr/si_lacs) to analyze the behavior of the three models on the long term on two large deep lakes: Lake Annecy (1966-2013) and Lake Geneva (1991-2013). This work on solely two lakes of similar characteristics prevents from reaching conclusions on the performance of the models in the long term.

We clarified the use of this period 1999-2016 for models performance assessment in section 4: Model performance and added the following (lines 305-309): The performance assessment was limited to the period of 1999-2016 due to the availability of water temperature data (in situ and satellite) during that specific timeframe. The scarcity of in situ water temperature measurements before 1999 applies to the entire set of lakes. However, it is important to note that long-term in situ water temperature data is available for a few large lakes, which was used to assess the performance of the three models (Prats & Danis, 2015).

*- Section 6: the authors should explain how they performed the sensitivity analysis and provide details about CSS. This should be anticipated in section 3.4.*
The sensitivity analysis of model parameters was conducted by calculating the CSS (Composite Scaled Sensitivities) statistic, which was done using the packages "CUSPY", and "pyemu" for which the functionalities are integrated in the package "ALAPROD". This is already mentioned in section 3.4 (now section 3.5, lines 260-262). We added more information (lines 267-280) and references about the CSS statistic to improve clarity as well as the equations for calculating CSS (Equations 13 and 14).

We reformulated as follows (lines 267-272): "Composite scaled sensitivities (*CSS*) indicate the quantity of information provided by each parameter and the sensitivity of the model to them (Ely,

**2006). The parameters with higher *CSS* values will have a greater impact on the resulting simulation compared to those with low *CSS* values. To determine the *CSS* values for each parameter, the Dimensionless Scaled Sensitivities (*DSS*) are used. *DSS* indicate how important an observation or how sensitive a simulated equivalent of an observation is in relation to the estimation of a parameter. Further information on these statistical measures is available in Hill (1998) and Poeter & Hill (1997)."**

*In their sensitivity analysis, the authors combine model parameters and forcing uncertainty (at_factor, sw_factor, MAAT). This looks a bit unusual: at_factor, sw_factor, MAAT are not model parameters.*
**Although *at_factor*, *sw_factor*, *MAAT* are not strictly speaking model parameters they can still influence greatly the outcome of the simulations as seen for example in the case of the "*at_factor*" which displays high CSS values. In fact, they give information on the sensitivity of model results to biases in forcing data. Reanalyses are not free from error, and often show biases over certain areas or types of terrain. We included the drivers along with the parameters in the sensitivity analysis to emphasize the potential biases and that they should be taken into account.**

*The analysis of Figure 4 is qualitative and should be improved: for which parameters are relationships between CSS and depth statistically significant? Are there statistically significant relationships also with other morphometric/geografical variables? The same considerations on the need to improve the analysis apply to Figure 2.*
**We agree that the provided analysis is mostly qualitative. Therefore, we used the Kendall's tau coefficient to determine the statistical association between (1) uncertainty values obtained with default simulations and lakes geomorphological characteristics (previously Figure 2, now Figure 3) and (2), CSS values of each model parameter and lakes geomorphological characteristics including maximal depth, volume, surface area, altitude and latitude (previously Figure 4, now Figure 5).**

**We also added tables to show the details about the obtained Kendall's tau coefficients and p-values for both uncertainty (Table 3 in section 5: Uncertainty analysis) and CSS (Table 4 in section 6: Sensitivity analysis) analyses in respect to lakes geomorphological characteristics.**

**We also added an appendix (Appendix A) in which supplementary figures were added to show the relationships between uncertainty ranges and lakes geomorphological variables (Figure A2) as well as the latter and CSS values (Figures A4-A7). Additionally, we added figures to Appendix A consisting of scatter plots (Figure A1) of lakes geomorphological characteristics as well as a correlation matrix (Figure A2) for the latter.**

**According to the obtained results, we reformulated sections 5 ("Uncertainty analysis") and 6 ("Sensitivity analysis"). The usage of the Kendall's tau as a statistic to determine these correlations was added in section 3.5 (lines 293-295). According to the obtained correlations, section 7 "Discussion and implications" (lines 396-405) was reformulated.**

*Minor comments:*

- *L147 and L152: please, clarify what you mean by "an exponential smoothing function"*
  **The smoothing function $f(*)$ is such that it gives greater weight to the nearest observations and the weights decrease exponentially. We added the equations (equations 3 & 4) related to this exponential smoothing function in the manuscript (lines 176-177) as presented in (Prats & Danis) 2019 (equations 6 & 7 in Prats & Danis, 2019) and clarified the use of it as mentioned above. For more clarification, we also added the equations (equations 6 & 7, lines 183-184) describing the exponential smoothing function $g(T_{e,i})$ of the epilimnion temperature (equations 9 & 10 in Prats & Danis, 2019) used for the hypolimnion module.**

- *L155: please, provide information about the ALAMODE model.*
  **The ALAMODE software suite is not a model, but a package integrating the functionalities of several models, packages and modules. We added an additional sub-section (3.1. The software suite ALAMODE) to section 3: "Data and methodology" in which we provide more details about ALAMODE and its utilization. (Danis, 2020) was used to reference ALAMODE.**

- *L158-L163: please, provide the basic information to understand how the default "parameterization presented in Prats & Danis (2019)" has been derived. "The expression for epilimnion ..." and "the parameterization of hypolimnion ...": do the author mean "the values of the parameters"? Are expression and/or parametrization (i.e., equation) different compared to eq. 1-3?*
  **By "expressions", we mean the derived equations used to estimate model parameters. For each parameter ($A, B, C, D, E, \alpha, \beta$) an equation was derived to estimate its value. For the epilimnion module, the equations were derived from robust regressions fitted between the parameter values and lakes geomorphological characteristics. For the hypolimnion module the expressions and parameter values of $\beta$ & $E$ were estimated as a function of depth and lake type, whereas $D$ was assigned a constant value. Landsat thermal data were used for the parametrization of the epilimnion module and in situ temperatures were used for the parametrization of the hypolimnion module.**

  **We already presented information about the "default" parametrization provided in Prats & Danis (2019) in section 3.2, previously section 3.1 (The OKP Lake Model description, lines 159-163). We reformulated the paragraph and added more information about this parametrization. We reformulated as follows (lines 187-196): "In ALAPROD, OKPLM can be run in two modes: the "default" mode where model parameter values for each lake are estimated using the parameterization presented in Prats & Danis (2019), and the "calibrated" mode where model parameters are calibrated individually for each lake by using in situ temperature measurements. The default parameterization was obtained by using the individually calibrated parameter values to fit appropriate expressions as a function of the characteristics of lakes. In the epilimnion module model parameter values are estimated (A, B, C, and α) based on lake characteristics (i.e., latitude, altitude, surface area, volume, and depth). These equations were determined using robust regressions and Landsat infrared data from 1999 to 2016 of French lakes to estimate surface temperatures (Prats et al., 2018). In contrast, for the hypolimnion module, parameter values (E and β) were derived as a function of lake depth and lake type using temperature profile data from 357 lakes; β can have a value of 1 (E > 0.95) or 0.13 (E ≤ 0.95). The parameter D was assigned a constant value of 0.51.**

  **We also add the equations (8, 9, 10, 11 & 12) related to the estimation of each parameter (already presented in Prats & Danis (2019)) (section 3.2: The OKP Lake Model description, lines 197-207).**

- *L188-195: I would restructure the paragraph anticipating lines 192-195 before lines 189-192. The authors say that S2M was used for simulating the water temperature in lakes situated at altitudes higher than 900 m, but in table 1, I see that some lakes at higher elevations have been run with SAFRAN.*
  **The paragraph was restructured according to your suggestions. In regards to the simulations of lakes situated at altitudes higher than 900 m with S2M data, this was done when possible. In order to use S2M meteorological data over each lake an extraction of certain topographic classes is necessary. These include elevation, aspect and slope, which represent the spatial variability over mountainous regions and are represented over areas called "massifs". On average, a massif corresponds to a mountainous region of about 1000 km² over which meteorological conditions are considered homogeneous at a given**

elevation range. Two types of S2M reanalysis simulations exist for each elevation range, one at flat terrain and the other with 8 aspects at 2 different slope angles (Vernay et al., 2022). For this study, this information (elevation, slope, aspect) was extracted from a Digital Elevation Model (BD Alti, IGN) for each lake over its drainage basin, combined into zones corresponding to S2M topographic classes. For some lakes ($n = 19$), it was not possible to use S2M data either because their drainage basins are not entirely part of a massif ($n = 1$), or because they are located in massifs that are not covered by the S2M reanalysis dataset ($n = 18$). The S2M renalaysis only covers the mountainous regions of Alps, Corsica and Pyrenees.

We clarified this in the manuscript (section 3.4: Lake simulations) and reformulated as follows (lines 249-251): "For some lakes, it was not possible to use S2M data, either because their drainage basins are not entirely part of a massif ($n = 1$), or because they are located in massifs that are not covered by the S2M reanalysis dataset ($n = 18$)."

We added some information about the extraction of S2M meteorological data (section 3.3: Input data) as follows (lines 218-228): "The S2M reanalysis uses a vertical resolution of 300 m and is the result of simulations performed over mountainous zones referred to as "massifs" and covering the French Alps, Pyrenees and Corsica mountainous regions. In order to use S2M meteorological data over each lake an extraction of certain topographic classes is necessary. These include elevation, aspect and slope, which represent the spatial variability over "massifs". On average, a massif corresponds to a mountainous region of about 1000 km2 over which meteorological conditions are considered homogeneous at a given elevation range. Two types of S2M reanalysis simulations exist for each elevation range, one at flat terrain and the other with 8 aspects at 2 different slope angles. For this study, this information (elevation, slope, aspect) was extracted from a Digital Elevation Model (BD Alti, IGN) for each lake over its drainage basin, combined into zones corresponding to S2M topographic classes. We considered a zero slope and average daily data for each study site."

- *Figures 1 and 2: I found the colour map difficult to appreciate, especially in the case of reservoirs (crosses).*
  We changed from a "cross" marker to represent reservoirs to a "diamond" marker and changed the colormap (colorblind friendly) in Figure 1 and Figure 2 (now Figure 3). The markers were updated in Figure 4 (now Figure 5) as well. If you have any other specific recommendations for color maps or markers style we would gladly integrate them in the figures.

- *L201: I am not sure what "initial assessment of the quality of OKPLM simulations" has been described in the previous section.*
  We agree; we wanted to reference the performance assessment that we describe in section 4 thus not the previous section. We have taken out this sentence as it is not appropriately positioned and may not be relevant to the information provided.

- *Section 3.4: the authors did not specify what objective function has been used (RMSE, NSE, MAE, other?). Is the range 0-1 for D, E and beta motivated by any physical/mathematical reasoning, or could it be wider?*
  The objective function used in the calibration was the squared sum of residuals. We have added this information to the article (line 265).

  We specified the use of Kendall's Tau coefficient for assessing the relationship between uncertainties and CSS in respect to lakes geomorphological characteristics in the section

**3.5 "Calibration, uncertainty and sensitivity analysis" (previously section 3.4) (lines 293-295).**

**Beta is a smoothing factor so typically its values range between 0 and 1. The parameters "*D*", "*E*" and "*β*", are expected to lie in the range 0-1 for mathematical and physical reasons. The parameter "*E*" value must be at most 1 to prevent temperature variations in the hypolimnion from being greater than those in the epilimnion. The parameter "*D*" multiplies the parameter "*A*", the average temperature of the epilimnion, to give an estimation of the average temperature of the hypolimnion (if there was no mixing). Reasonably, its value should not be greater than 1 or less than 0.**

**We added the following sentence for more clarification (lines 287-289): "The parameters *D*, *E* and *β*, are expected to lie in the range 0-1 for mathematical and physical reasons. However, their respective values are highly interdependent and are difficult to identify."**

- *Table 2: in the caption the authors mention a tilde, but they use a circumflex accent.*
  **We corrected this in the Table 2 caption (from "tilde" to "circumflex accent").**

- *L226: I think the authors should refer to https://hess.copernicus.org/articles/17/3323/2013/ for air2water.*
  **We referred to the air2water model as suggested (line 300) and we left the reference Toffolon et al., 2014 since it references the 4-parameter version of the air2water model used in Prats & Danis (2019).**

- *Figure 3: please use capital letters for the parameters in the x-axis, as in the text. The font should be adjusted. mat should be modified into MAAT (also at line 265).*
  **Figure 3 (now Figure 4) was modified according to your suggestions.**

✦ **Additional author comments:**
**The LakeTSim dataset comprises simulations of reservoirs and artificial lakes for which the dates of initial filling are not always known. However, for the majority of these lakes (reservoirs and other artificial lakes) we were able to extract the initial filling years from https://www.barrages-cfbr.eu and the PLAN_DEAU database. We show this data in Table S1 which we added in the Supplement. This supplementary dataset lacks information for some reservoirs, all gravel pits and other artificial lakes. Note that we did not consider the lakes initial filling dates in the version of the paper we first submitted. However, we think that it is important to show this information especially for future users of the LakeTSim dataset. Therefore, we added an additional section (section 8. Data usage) to present this information and provide general recommendations about the use of the water temperature simulations dataset. We also added an additional figure (Figure A8 in Appendix A) to show the distribution of the initial filling years for the reservoirs and other artificial lakes present in the LakeTSim dataset.**

**We also added a new figure (Figure 2 in section 3.4. Lake simulations) with the objective to present the LakeTSim data. We thought that a visual of what the dataset is (daily simulations of epi- and hypolimnion temperature for 401 lakes over 60 years, with uncertainties) can help users to assimilate the data.**

**Finally, we added the uncertainty data of epilimnion and hypolimnion temperature simulations for each lake to the LakeTSim dataset for publication. While this was not requested by the reviewers, we think this data can be useful for users who wish to assess the quality of the simulations and/or use the data from one lake only. The structure of the LakeTSim folders was updated to accommodate this addition.**

[Figure]

[Figure]

**References:**

[revised manuscript text omitted]

---

## Referee Report (RR1)

**A long-term dataset of simulated epilimnion and hypolimnion temperatures in 401 lakes (1959-2020)**

The manuscript presents a dataset of simulated lake water temperature of 401 French lakes using a modelling approach. The manuscript was already in a good shape but I feel that after this round of revision the description of calibrated or not versions (and their comparison) greatly improved.

I think that the presented dataset is relevant to leverage lake research in the context of climate change. The manuscript it is overall well written so I only have some minor issues listed below.

**Minor issues:**

- L194 – "surface temperature" should be "mean surface temperature" right? Because potentially Landsat give the spatially distributed information.
- Figure 1 – The legend report: Reservoirs, natural, artificial (others) and gravel pit. I am curious what kind or lakes have been grouped under artificial, could you please add an example in the text where you mention this distinction for the first time? That would be useful to understand.
- L367 – add a space before parenthesis.
- L462 – I suggest modifying the sentence by adding: "We compiled the initial temporal gap filling for 282 of…."

---

## Author Response (AR2)

**Response letter**

**Preprint title (ESSD-2022-457):** A long-term dataset of simulated epilimnion and hypolimnion temperatures in 401 French lakes (1959-2020).

**Authors:** Najwa Sharaf, Jordi Prats, Nathalie Reynaud, Thierry Tormos, Rosalie Bruel, Tiphaine Peroux, Pierre-Alain Danis.

Reviewers comments are in *italic* and authors comments are in **bold**.

**On behalf of my co-authors, I would like to thank the reviewers for taking the time to review our manuscript for the second time. We carefully considered all their suggestions and made the necessary revisions to the manuscript. In the sections below, we provide responses to each of the issues raised by the reviewers.**

**🔅 Reviewer 1**

*The manuscript presents a dataset of simulated lake water temperature of 401 French lakes using a modelling approach. The manuscript was already in a good shape but I feel that after this round of revision the description of calibrated or not versions (and their comparison) greatly improved.*
*I think that the presented dataset is relevant to leverage lake research in the context of climate change. The manuscript it is overall well written so I only have some minor issues listed below.*

*Minor issues:*
- *L194 – "surface temperature" should be "mean surface temperature" right? Because potentially Landsat give the spatially distributed information.*
  - **Yes, Landsat infrared data (skin temperatures) were used to estimate mean surface temperatures according to (Prats et al., 2018).**
  - **Within a Landsat image, skin temperatures were averaged spatially over water pixels. According to Prats and Danis, (2019), median skin temperatures were used as an estimation of mean surface temperatures.**
  - **We reformulated the sentence as follows (L193-195): These equations were determined using robust regressions and Landsat infrared data (median skin temperatures) from 1999 to 2016 of French lakes to estimate mean surface temperatures (Prats et al., 2018).**
- *Figure 1 – The legend report: Reservoirs, natural, artificial (others) and gravel pit. I am curious what kind or lakes have been grouped under artificial, could you please add an example in the text where you mention this distinction for the first time? That would be useful to understand.*
  - **The category of 'other' artificial lakes (n = 38) primarily comprises ponds and quarry lakes. This information has already been included in section 3.4 "Lake simulations" (L243) and section 8 "Data usage" (L483). For this revision, we added the same information in the legend of Figure 1 and the rest of the figures where the category "Artificial (other)" is mentioned (Figures 3, 5, A3, A4, A5, A6, A7, & A8).**

[Figure]

- – **We specifically added the following sentence to the legends: The "other" artificial lakes consist of ponds and quarry lakes.**

    - – **In Figure A8 (Appendix A) we modified the legend as follows: "Distribution of initial filling years for lakes (e.g., reservoirs, gravel pits, ponds and quarry lakes) of the LakeTSim dataset."**

- *L367 – add a space before parenthesis.*
    - – **Space added (L389).**

- *L462 – I suggest modifying the sentence by adding: "We compiled the initial temporal gap filling for 282 of…."*
    - – **We modified the sentence as follows (L489): "We compiled the initial temporal gap filling related to the initial filling years for 282 of these 347 non-natural lakes (269 reservoirs and 13 artificial lakes, Figure A8 in Appendix A) in Table S1 (see Supplement) to be used as a companion dataset to LakeTSim."**

**✚ Reviewer 2**

*I appreciate the changes included into the manuscript, especially concerning the uncertainty analysis (both methods and results).*

*Please, find below a couple of specific comments.*

- *As for the authors' reply to the first main comment: please note that the 8-, 6-, and 4-parameter versions of the air2water model were already presented in Piccolroaz et al (2013), while Toffolon et al (2014) and Piccolroaz (2016) revised the definition of the parameters. Please, specify that you used the 4-parameter version of the model.*
    - – **We specified the use of the 4-parameter version of the air2water model in section 4 "Model performance", L320.**

- *Note that the parameters that you used were those obtained in Toffolon et al (2014) and recalled in Piccolroaz (2016) after having applied the model to 14 temperate lakes around the globe. In this respect, I do not think that it is fair to compare the OKPLM model run with the "default" parameter values given by the parameterization in Prats & Danis (2019) with air2water run with the parameters in Toffolon et al (2014) and Piccolroaz (2016). In fact: in the first case, the parameters have been obtained for the same/a similar dataset (French lakes) while in the second case they refer to lakes from around the globe. Being the air2water model data-driven, one cannot assume the parameters in Toffolon et al (2014) and Piccolroaz (2016) as "default parameters" and use these parameters for other applications, especially if the air temperature (the forcing) comes from a different source.*
    - – **According to (Piccolroaz, 2016), the 4-parameter version of the air2water model (with calibrated parameters) shows acceptable performance until the percentage of missing data reaches about 97% (i.e., when data are available at about monthly resolution, on average) which is the case for most water bodies in the LakeSST dataset used in Prats and Danis, (2019). Hence the choice to apply the 4-parameter version of the air2water model in Prats and Danis, (2019) using the parametrization presented in Toffolon et al. (2014) and derived from a set of 14 temperate lakes with different geomorphological characteristics.**
    - – **In order to clarify this we added the following in section 4 "Model performance" (L331-336): The air2water parameter values were obtained as a function of lake depth from the parametrization presented in Toffolon et al. (2014), based on data from 14 lakes around the globe. In this case, the air2water model parameters were**

[Figure]

not calibrated due to the fact that the percentage of missing data within the LakeSST dataset employed in Prats & Danis (2019) exceeded 97% for most lakes. Beyond this threshold of 97% missing data, the performance of the calibrated 4-parameter version of the air2water model was found to be unsatisfactory (Piccolroaz, 2016).

- *Equation(14): please, explain how the weights w_i are assigned.*
    - The weights ($w_i$) are calculated based on the inverse of the variance-covariance matrix of the observation errors.
    - We added this information on L289, section 3.5 "Calibration, uncertainty and sensitivity analysis".
- *According to the authors' response, I would stress that they "included the drivers along with the parameters in the sensitivity analysis to emphasize the potential biases and that they should be taken into account".*
    - Thank you for highlighting our response in regards to the inclusion of forcing parameters in the sensitivity analysis, we still maintain this perspective. We think that this indeed underscores the importance of considering potential forcing biases in our analysis.
    - We added this information to the manuscript, Section 3.5 "Calibration, uncertainty and sensitivity analysis", as follows (L271-274): "In addition to model parameters, sensitivity analysis was extended to encompass forcing parameters (*MAAT, at_factor, sw_factor*) as they provide information about the degree of sensitivity exhibited by model parameters in response to biases in the forcing data."

**Additional author comments**

    - We added some minor modifications to the manuscript (highlighted in the marked version of the submitted manuscript).
    - We added some details to the section 3.4 "Lake simulations" to clarify more the difference between the usage of "default" and "calibrated" model parameters (L247-255).

**References**

Piccolroaz, S.: Prediction of lake surface temperature using the air2water model: Guidelines, challenges, and future perspectives, Adv. Oceanogr. Limnol., 7, 36–50, https://doi.org/10.4081/aiol.2016.5791, 2016.

Prats, J. and Danis, P. A.: An epilimnion and hypolimnion temperature model based on air temperature and lake characteristics, Knowl. Manag. Aquat. Ecosyst., 8, https://doi.org/10.1051/kmae/2019001, 2019.

Prats, J., Reynaud, N., Rebière, D., Peroux, T., Tormos, T., and Danis, P. A.: LakeSST: Lake Skin Surface Temperature in French inland water bodies for 1999-2016 from Landsat archives, Earth Syst. Sci. Data, 10, 727–743, https://doi.org/10.5194/essd-10-727-2018, 2018.

Toffolon, M., Piccolroaz, S., Majone, B., Soja, A. M., Peeters, F., Schmid, M., and Wüest, A.: Prediction of surface temperature in lakes with different morphology using air temperature, Limnol. Oceanogr., 59, 2185–2202, https://doi.org/10.4319/lo.2014.59.6.2185, 2014.

---

## Author Response (AR3)

**Response letter**

**Preprint title (ESSD-2022-457):** A long-term dataset of simulated epilimnion and hypolimnion temperatures in 401 French lakes (1959-2020).

**Authors:** Najwa Sharaf, Jordi Prats, Nathalie Reynaud, Thierry Tormos, Rosalie Bruel, Tiphaine Peroux, Pierre-Alain Danis.

Editor's comments are in *italic* and authors comments are in **bold**.

*Dear authors,*

*thank you very much for all your efforts w.r.t. the revision of your manuscript. Both remaining reviewers only had some minor comments, that were carefully addressed in your last version.*

*Hence, I am happy to inform you that your manuscript is now accepted for publication in ESSD!*

*I would also like to concur with the opinion of the two reviewers and emphasize again that your dataset and the corresponding manuscript are of high value for the broader scientific community. Particularly due to the length of the time-series of more than six decades, it allows for assessing long-term changes in the studied lakes which, obviously, is of high scientific, but also societal, relevance.*

*If you have any further questions, please do not hesitate to contact me.*

*Best regards and, again, thank you very much,*

*Christof Lorenz*

*Additional private note (visible to authors and reviewers only):*

*Dear authors,*

*I have only one last comment... ESSD requires the DOI / reference of the corresponding dataset to be included in the abstract. Please have a look at the "Manuscript composition" under https://www.earth-system-science-data.net/submission.html" and follow these guidelines. You can also check some other published manuscripts for example wordings.*

*If you need help or if you have any questions, please let me know!*

*Best regards,*
*Christof Lorenz*

[Figure]

**Dear Mr. Lorenz,**

**On behalf of my co-authors, I would like to thank you for your response, for handling our manuscript and getting it reviewed as well as for accepting it for publication.**

**According to your comment, we added the doi/reference relative to our dataset (LakeTSim) in the abstract.**
**Additionally we did minor corrections to the text in sections 3.3 and 3.4 as follows:**

**L238-240 (section 3.3), we replaced:**
**"In situ temperature profiles were extracted for 170 lakes over different depths."**
**By:**
**"In situ temperature profiles were extracted from the RCS/RCO (Réseau de Contrôle de Surveillance/Réseau de Contrôle Opérationel, French networks for WFD) monitoring network for 170 lakes over different depths."**

**L 250-251 (section 3.4), we replaced:**
**"Specifically, "calibrated" model parameters were adopted when in situ temperature profiles along the water columns were available from the RCS/RCO (Réseau de Contrôle de Surveillance/Réseau de Contrôle Opérationel, French networks for WFD) monitoring;"**

**By:**
**"Specifically, "calibrated" model parameters were adopted when in situ temperature profiles along the water columns were available from the RCS/RCO monitoring network".**

**Again, thank you very much.**

**Best regards,**
**Najwa SHARAF**